# Applications of Cytokinins in Horticultural Fruit Crops: Trends and Future Prospects

**DOI:** 10.3390/biom10091222

**Published:** 2020-08-22

**Authors:** Adeyemi O. Aremu, Olaniyi A. Fawole, Nokwanda P. Makunga, Nqobile A. Masondo, Mack Moyo, Nana M. D. Buthelezi, Stephen O. Amoo, Lukáš Spíchal, Karel Doležal

**Affiliations:** 1Indigenous Knowledge Systems Centre, Faculty of Natural and Agricultural Sciences, North-West University, Private Bag X2046, Mmabatho 2745, South Africa; AmooS@arc.agric.za; 2Food Security and Safety Niche Area, Faculty of Natural and Agricultural Sciences, North-West University, Private Bag X2046, Mmabatho 2745, South Africa; 3Postharvest Research Laboratory, Department of Botany and Plant Biotechnology, Faculty of Science, University of Johannesburg, Auckland Park Kingsway Campus, P.O. Box 524, Auckland Park 2006, South Africa; butheleziduduzile@gmail.com; 4Department of Botany and Zoology, Stellenbosch University, Private Bag X1, Matieland 7602, South Africa; Masondo@sun.ac.za; 5Department of Horticulture, Faculty of Applied Sciences, Durban University of Technology, P.O. Box 1334, Durban 4000, South Africa; MackM@dut.ac.za; 6Agricultural Research Council, Roodeplaat Vegetable and Ornamental Plants, Private Bag X293, Pretoria 0001, South Africa; 7Department of Chemical Biology and Genetics, Centre of the Region Haná for Biotechnological and Agricultural Research, Faculty of Science, Palacký University, Šlechtitelů 27, CZ-783 71 Olomouc, Czech Republic; lukas.spichal@gmail.com (L.S.); karel.dolezal@upol.cz (K.D.); 8Laboratory of Growth Regulators, Faculty of Science, Palacký University & Institute of Experimental Botany AS CR, Šlechtitelů 11, CZ-783 71 Olomouc, Czech Republic

**Keywords:** abiotic stress, biotechnology, food security, micropropagation, shoot proliferation, somatic embryogenesis, phytohormones, plant growth regulators, postharvest, quality attributes

## Abstract

Cytokinins (CKs) are a chemically diverse class of plant growth regulators, exhibiting wide-ranging actions on plant growth and development, hence their exploitation in agriculture for crop improvement and management. Their coordinated regulatory effects and cross-talk interactions with other phytohormones and signaling networks are highly sophisticated, eliciting and controlling varied biological processes at the cellular to organismal levels. In this review, we briefly introduce the mode of action and general molecular biological effects of naturally occurring CKs before highlighting the great variability in the response of fruit crops to CK-based innovations. We present a comprehensive compilation of research linked to the application of CKs in non-model crop species in different phases of fruit production and management. By doing so, it is clear that the effects of CKs on fruit set, development, maturation, and ripening are not necessarily generic, even for cultivars within the same species, illustrating the magnitude of yet unknown intricate biochemical and genetic mechanisms regulating these processes in different fruit crops. Current approaches using genomic-to-metabolomic analysis are providing new insights into the *in planta* mechanisms of CKs, pinpointing the underlying CK-derived actions that may serve as potential targets for improving crop-specific traits and the development of new solutions for the preharvest and postharvest management of fruit crops. Where information is available, CK molecular biology is discussed in the context of its present and future implications in the applications of CKs to fruits of horticultural significance.

## 1. Introduction

Cytokinins (CKs) are a unique class of plant growth regulators (PGRs) with a long and interesting history. Their existence as compounds capable of inducing cell division in cultured plant tissues was first documented more than 100 years ago [1]. With the discovery of an increasing number of compounds with CK-like actions in plants even to date, CKs are thus broadly grouped as natural (purine-based molecules, which are either isoprenoid or aromatic CKs) or synthetic CKs, which are urea-based [2]. Figure 1 shows the structural configurations of some existing natural and synthetic CKs. These CKs are considered to possess potential influence throughout the entire course of a plant’s life from embryogenesis until death in both lower and higher plants, as evidenced in the diverse physiological and biochemical functions during the life cycle of the different organisms [2,3,4]. They are involved directly or indirectly in different plant physiological processes such as the regulation of seed germination, shoot elongation and proliferation, induction of flowering, fruiting and seed set, and senescence [5,6,7,8,9,10]. Particularly, their roles in fruit set, delay of senescence processes—including fruit ripening and defoliation [11], which are concomitant with the release of buds from apical dominance [12], remain fundamental to the successful production of many horticultural fruit crops. Coupled with the development of genetically improved crop varieties and the application of improved agronomic practices, the use of PGRs including CKs has contributed positively to the green revolution and subsequent increase in agricultural productivity globally [13]. However, fundamental knowledge of the diverse roles of CKs in plants remains fragmented, and there is greater scope to deepen our knowledge of how CKs function and regulate cellular mechanisms that control plant growth and development. This knowledge will enable greater exploitation and application of CKs in horticultural fruit production. Recently, Koprna et al. [14] highlighted the potential of CKs as agrochemicals in pot and field experiments as they improve the growth dynamics and yields of a wide range of plants, including horticultural fruit crops.

With more than 80 commonly known species of horticultural fruit crops available, their relevance to offset food and nutrition security concerns among the ever-increasing global population cannot be overemphasized. For centuries, horticultural fruit crops have been cultivated (mainly via conventional methods) as important dietary foods serving as the major sources of vitamins, antioxidants, and fibers for human needs [15,16]. As an indication of their economic and commercial values, the global production of the major fruit crops, including banana, apple, orange and mango, has witnessed a consistent and dramatic increase according to the Food and Agriculture Organization (FAO). Statistics from the FAO show that between 2000 and 2017, the production of mangoes, mangosteens, and guavas rose from 20 to over 40 million tonnes while banana production experienced a compound annual growth rate of 3.2% over the same period (http://www.fao.org/statistics/en/). Figures for banana cultivation were on a record high in 2017, reaching 114 million tonnes from 67 million tonnes in 2000. Other fruit crops such as oranges, apples, and grapes also showed positive trajectories in terms of their production, even though their incremental trends did not surpass that of banana (http://www.fao.org/statistics/en/). While these increases remain laudable, more effort to stimulate higher yield potential and the stability of the major fruit crops are needed to feed a world population that is predicted to reach 8 billion by 2025 [13].

The propagation of many fruit crops has intrinsic challenges such as low germination rate, heterozygosity of seeds, and prolonged juvenile phase, which hamper efficient and rapid growth [17]. Together with the changing climate, biotic and abiotic stresses can significantly influence productivity in major fruit crops [18,19,20,21,22,23]. In recent times, different strategies including genetic modification [19,24,25,26], encapsulation technology [20], photo-biotechnology [27], and the manipulation of phytohormone balances with compounds such as nitric oxide (NO) [28] and 1-methylcyclopropene (1-MCP) [29] have been explored to mitigate biotic and abiotic stresses. Furthermore, the systematic application of biostimulants, particularly plant growth-promoting rhizobacteria (PGPR) and mycorrhizal fungi have been demonstrated to hold the potential to mitigate biotic and abiotic stresses as well as boost fruit crop production [18,30]. In addition to these approaches, the diverse roles of PGRs, especially CKs, offer a potential avenue that requires more detailed attention [31,32,33,34,35,36]. As an example, Zalabák et al. [22] postulated that the genetic engineering of CK metabolism may offer greater potential to improve the agricultural traits of crops. In response to environmental cues, physiological and genome-wide microarray studies indicate an existing relationship with CK levels in planta [32]. In addition to increasing evidence of CKs’ influence in alleviating biotic and abiotic stresses, CKs play an important role in horticultural crop production where their application influences the morphological structure and nutrient content, as well as facilitates harvesting and the overall yield in a number of fruit crops [14,37]. Thus, in this review, we highlight and critically explore the potential of CKs in the propagation, growth, and general physiology with specific reference to some fruit crops. In the past three decades, the advent of molecular biology, genetic engineering, and exploitation of mutant technologies in various model plant species has led to a better understanding of the molecular mechanisms of CKs. Major breakthroughs in the 1990s led to the discovery of the CK signaling circuit networks that partly explain the diverse roles of CKs throughout a plant’s life cycle in molecular, cellular, and developmental contexts. Some of this research, mainly conducted in the non-horticultural *Arabidopsis* species (*Arabidopsis thaliana*) as a model, has been comprehensively reviewed in the works of:(a)Kakimoto [38], describing the perception and signal transduction mechanisms involving CK receptors in plants based on the molecular work conducted in the 1990s;(b)Hwang et al. [39], where CK–auxin relationships controlling early embryogenesis and organ differentiation and development are explained. The authors highlighted studies conducted in *Lotus japonicus* and *Medicago truncatula* that have led to the acknowledgement of the importance of CKs in nodule formation. Furthermore, the impacts of CK circuits in biotic and abiotic stress responses and regulation of senescence by CKs were critically described;(c)Steklov et al. [40], who compared the structural configuration of CK receptors and their phylogenetic relatedness across species including horticultural crops such as orange, apple, tomato, and grape.

This paper is not a comprehensive review of the mechanisms of CK in plants; nonetheless, we briefly describe the available current information and new insights into the molecular mechanisms and modes of action of CKs with particular reference to horticultural fruit crops. The paper details the impacts of CK application in pre and postharvest management practices of horticultural fruit crops. Based on existing data, we identify gaps in knowledge and recommend potential ways to explore the value of CKs in horticultural fruit crops. Finally, we highlight the possibilities in exploring horticultural fruit crops as new models to provide a better understanding of the broader functioning of CKs and their regulatory control in horticultural fruit crops. Tomato, a prominent model organism for scientific studies, is excluded due to the extensive existing literature focusing on different aspects of the plant [15,27,41,42,43].

## 2. An Overview of the Mode of Action of Cytokinins in Plants

The functions of CKs in plants involve complex coordination through a diverse network of cross-talk mechanisms that results in the regulation of numerous physiological processes, namely, axillary shoot branching, the release of apical dominance, root meristematic cell patterning, and the production of lateral roots [14]. More often, the processes that are mutually controlled by CKs and other PGRs are defined scientifically as being antagonistic or agonistic. However, this viewpoint is regarded as being overly simplistic. For example, Schaller et al. [44] emphasized the importance of revisiting these definitions. Our current understanding of the relationship of these PGRs lends itself to their roles in plants being redefined so that they are more reflective of their actual actions. Thus, it may be best to define their roles as being ‘complementary and dynamic’ as PGRs function synergistically, antagonistically, or additively to bring a desired result. For example, CKs together with auxins are important for stem cell differentiation and the activities of CKs and auxins at both the shoot apical meristem (SAM) and root apical meristem (RAM) regions, and their cross-talk interplay are excellently described by Schaller et al. [44].

### 2.1. Metabolic Regulation of Cytokinin Activity

The metabolic production and control (biosynthesis, inter-conversions, and degradation) of CK homeostasis involve a wide range of enzymes [12,45]. Particularly, isopentenyltransferase (*IPT*) is an important enzyme involved in the first and rate-limiting step in CK biosynthesis that entails the transfer of an isoprenoid moiety to the *N*^6^ position of the adenine nucleotide [22,45]. An additional enzyme involved in the modification of CKs at the adenine part of the molecule was discovered in 2007. Evidence from the study by Kurakawa et al. [46] revealed the existence of a specific phosphoribohydrolase (designated as Lonely Guy; *LOG*) in rice. The *LOG* enzyme is responsible for the cleavage of ribose 5’-monophosphate from the CK nucleotides to form biologically active CK-free bases in one enzymatic step [22]. On the other hand, cytokinin oxidase/dehydrogenase (*CKX*) is central to the catabolism of CKs, where an irreversible cleavage of the CKs occurs, and the presence of auxins positively regulates this enzyme. Cytokinin oxidase/dehydrogenase is under a positive auxin regulation, leading to the regulated synthesis of CKs in plants and associated responses. The CK biosynthetic genes belong to a gene family that is developmentally and spatially regulated in its expression in plant cells [12,22].

Glucosyltransferases and xylosyl transferases catalyze *O*-glucosylation, *N*-glucosylation, and *O*-xylosylation events, leading to the production of various CK conjugates whose full function remains to be completely characterized [47,48]. For instance, a recent evidence revealed the metabolic reactivation of *trans*-zeatin (*t*Z) *N*-glucosides (*N^7^* and *N^9^* positions) in *Arabidopsis thaliana*, which is contrary to the previously-held hypothesis that *N*-glucosylation irreversibly inactivates CKs [45]. Many of these enzymes involved in CK metabolism were discovered mainly in the 1990s through to the 2000s. The uridine diphosphate glycosyltransferases (UGTs) are now known to deactivate CKs such that the regulation of CKs in plants is precise during distinct developmental phases and in response to environmental conditions throughout the plant’s life [49]. Environmental factors, both abiotic and biotic, as well as endogenous inputs, tightly regulate the synthesis and degradation of CKs, generally, in plants [50].

### 2.2. Molecular Aspect of Cytokinin Actions

Molecular genetic approaches have been useful in unravelling the major sensing and signaling roles linked to CKs [12]. The CK receptors are of a histidine kinase (HK) nature with autophosphorylation events being important as part of the signaling transduction pathways that ultimately lead to the negative and positive induction of CK-controlled gene expression [39]. The CK signal pathway in plants uses a basic phosphorelay two-component system (firstly described in bacteria) which revolves around four sequential phosphorylation steps that alternate between histidine and aspartate residues, where a conserved CK-binding domain, Cyclases/Histidine kinases Associated Sensory Extracellular (CHASE), has an extracytosolic location [39]. The HK receptors are localized on endoplasmic reticulum (ER) membrane, and the CHASE domain lies in the direction of the ER, leading to the hypothesis that the in planta binding of CKs is in the lumen of the ER [47]. These CK receptors are part of a large family of transmembrane HK sensors with three main evolutionary branches in plants, which is evident through the application of various bioinformatics tools [40]. The cytokinin response factor (*CRF*) gene families are known to control cotyledon and leaf development. Although the *CRF* genes, belonging to the family of AP2/ERF transcription factors, were first characterized in *Arabidopsis*, they are found in all land plants [51]. The tomato-specific CRFs, termed *SICRF* genes, responds to CKs by controlling the development of leaf primordial and root tips, and they occur as two distinct clades [52]. The review by Cortleven et al. [53] highlights the importance of CK mutants in uncovering the signaling mechanisms and biosynthesis steps involved in the in planta production of natural CKs. For example, *LOG* enzymes catalyse the reaction steps that increase the metabolic pool of CKs such as isopentenyladenine (iP) and *t*Z in plant tissues [10,12]. The nuclear-localized type B response regulators (RRB or type B ARR) are transcription factors of the CK signaling pathway that CK targeted for gene expression [12,53]. Through a negative feedback loop, the other regulators, type A RRs (RRA), indirectly control the induction of the CK-responsive genes that are in fact targets of the type B RRBs [53].

As a result of the benefits associated with transgenic or genome engineering, desired traits can be manipulated in different horticultural fruit crops (Table 1), and this has largely been spurred on by accumulating new information on the molecular biological effects of CKs in plants. For instance, genes related to specific CKs such as CPPU (*N*-(2-chloro-4-pyridyl)-*N*´-phenylurea) and BA (*N*^6^-benzyladenine) were recently identified in horticultural fruits. Following the treatment of pear fruitlet with 30 mg/L CPPU, the *B-PpRR* genes potentially influenced fruit development, bud dormancy, and light/hormone-induced anthocyanin accumulation [54]. The study by Ni et al. [54] indicated that CKs have the potential to stimulate the accumulation of anthocyanin in pear. Similarly, the upregulated expression of the *LDOX* gene contributed to the induction of anthocyanin content in strawberry treated with varying concentrations of CPPU [55]. Apart from the impact of CKs on specialized (secondary) metabolites [54,55], central (primary) metabolites, especially the carbohydrate content in fruits, may be indirectly influenced by CKs, as shown in kiwifruit [56] and strawberry [55]. Dipping application of kiwifruits in 10 mg/L CPPU significantly influenced the soluble carbohydrate component of the fruit osmotic pressure [56]. In apple, evidence of the expression of different genes related to CK activities was shown during axillary bud development [57] and flowering [58,59]. The expression of these CK-related genes was postulated to be essential for the postharvest storage of horticultural fruits, including strawberry [55].

Recently, genome editing in fruit crops by CRISPR/Cas9 has emerged as an alternative approach to mitigate time-consuming conventional breeding programmes [26,60,61]. Since the first studies in tomato and citrus-producing stable transgenic lines, the CRISPR/Cas9 technology has been applied to an increasing list of fruit crops including kiwifruit, banana, strawberry, papaya, and ground berry [62]. Genome-wide expression analysis data are largely lacking for many aspects linked to the developmental biology of fruit crops. Available information is mainly for the fruit biology of horticultural crops and genes linked to defense responses but not necessarily linked to CK responsiveness [51]. Despite increasing efforts, the molecular mechanisms underlying the role of CKs in pre- and postharvest quality performance of horticultural fruits are yet to be fully elucidated, and such information may be critical for the utilisation of modern technologies for fruit crop improvement.

## 3. Effects of Cytokinins in Different Phases of Horticultural Fruit Crops

The multifaceted functions of CKs in plants have led to their applications in commercial horticulture, notably micropropagation (Table 2 and Table 3). In addition, the potential of CKs as a viable tool for the manipulation of critical aspects of plant growth and development such as fruit size and quality for maintaining quality aspects of agricultural produce has become more apparent, and this intermediary phase brings the produce in closer connection to markets and consumers. To demonstrate the importance of CKs in horticultural fruit crops, their diverse roles in plant growth and development are discussed in detail in the subsequent sections and summarized in Figure 2.

### 3.1. Micropropagation Phase

Micropropagation remains a fundamental biotechnology approach that is capable of producing clonal propagules, which are pathogen- and disease-free due to the aseptic nature of the technology, within the shortest possible time. Plant propagation using tissue culture technology has become a routine and sometimes the sole-practical procedure for the production of high-quality clones, including fruit trees, in commercial horticultural systems [17,74,75]. A highly efficient plant regeneration protocol is often a prerequisite regardless of the type of biotechnological approach used for the improvement of the fruit crops. Pioneer studies in in vitro plant manipulation demonstrated the importance of CKs as well as their interactions with auxins on morphogenesis [44], which has led to a multitude of tissue culture-based regeneration regimes in fruit species of economic value.

#### 3.1.1. Mass Propagation of Horticultural Fruit Crops

Efforts to mass propagate different horticultural fruit species have been actively pursued globally (Table 2 and Table 3). For instance, in apple, extensive research has been conducted to standardize tissue culture protocols with significant emphasis on the type of CK [75,76]. Researchers have recognized the importance of CKs in clonal propagation to supply uniform propagules to the growing horticultural fruit industry. Over the past few decades, there has been significant progress and advances in the use of CKs for the clonal mass propagation of elite genotypes of horticultural fruit trees via organogenesis and somatic embryogenesis in both commercial and research laboratories. The importance and impact of CKs in micropropagation have been widely reported and are well-entrenched in the field of horticulture [77]. It is also highly relevant to reduce the relatively long juvenile phase associated with perennial fruit tree species. Somatic embryogenesis has emerged as a better alternative to organogenesis in the mass propagation of some fruit trees, due to the in vitro recalcitrance of explants from mature phase selections [78,79,80,81,82,83]. However, irrespective of the propagation method, CKs are often the main determinants of successful plant regeneration.

#### 3.1.2. Influence of Cytokinins in Micropropagation of Horticultural Fruit Crops

There are numerous reports on the optimization of CK types and concentrations in organogenesis and somatic embryogenesis protocols for many horticultural species, including the commercially cultivated fruits such as *Citrus* spp., *Malus* spp., *Litchi chinensis*, *Psidium guajava*, *Musa* spp., *Passoflora edulis*, *Punica granatum*, *Vitis vinifera,* and *Carica papaya* ( Table 2 and Table 3). Aspects of mass propagation in important horticultural fruit crops using organogenesis and/or somatic embryogenesis have been summarised in excellent plant-specific reviews for *Psidium guajava* [80,82], *Punica grantum* [81,155,156], and *Malus* spp. [75]. Various explants such as petioles, leaves, shoot meristems, seeds, cotyledons, anthers, filaments, pistils, nucellar, endosperms, inner integuments, and protoplasts can be used for the induction of somatic embryos. However, immature zygotic embryos represent the most desired source of embryogenic cells in most established somatic embryogenesis protocols—for example, in *Citrus* spp., *Litchi chinensis*, *Musa* spp., *Persea americana,* and *Psidium gujava* (Table 2). The effectiveness of immature zygotic embryos for the in vitro mass propagation of many recalcitrant fruit trees is dependent on the presence of pre-embryogenic determined cells, which have high embryogenic competence and can be easily induced into embryogenic masses by combinations of CKs, auxins (mainly 2,4-D and picloram), and other PGRs, including gibberellin (GA₃) [80]. The influence of CKs in plant propagation via somatic embryogenesis was reported in several fruit species, notably *Citrus* spp. [78,93,157], *Psidium guajava* [158,159,160], *Prunus* spp. [161,162], and *Vitis* spp. [163]. Furthermore, efficient plant regeneration has become an integral component of molecular genetic techniques of plant improvement, which form the mainstay in the production of transgenic plants [74].

Besides the vital factors such as type of explant, the season of explants’ collection, endogenous CK concentration, as well as the age and genotype of the stock plant, the success of any plant propagation system using plant tissue culture rests almost exclusively on the applied PGRs, especially the type and concentration of CK. The response of different horticultural plants to CK type and concentration is as widely varied as the plant species themselves (Table 2 and Table 3). Benzyladenine is unrivalled as the main CK used in the micropropagation of horticultural fruits crops through both organogenesis and somatic embryogenesis. In mass propagation through organogenesis, other commonly used CKs include KIN, zeatin, zeatin riboside (ZR), *N*⁶-(2-isopentyl) adenine, *N*-(2-chloro-4-pyridyl)-*N*-phenylurea, thidiazuron (TDZ), and most recently, the *meta*-topolins (Table 2). On the other hand, TDZ and KIN are the predominant alternatives to BA for somatic embryo induction, maturation, and plant recovery in most fruit trees of economic importance (Table 2). By and large, BA together with 2,4-D [164] remain the most commonly used synergistic CK–auxin combination in the plant propagation of important fruit tree species such as banana, apple, grapes, litchi, sweet orange, passion fruit, avocado, and guava through somatic embryogenesis (Table 2). Moreover, plant recovery stages depend primarily on a CK–GA₃ combination [135,139,145,146,147]. However, despite early promising results of *N*^6^-(3-hydroxybenzyl) adenine (*meta*-topolin, *m*T) as a possible alternative to BA in plant tissue culture [165,166] and its subsequent application in the organogenesis of numerous plant species [167], there is a dearth of information on the use of *meta*-topolins in the somatic embryogenesis of fruit crops. This paucity of knowledge presents an interesting area of research in the bid to increase the efficiency and effectiveness of CKs in somatic embryo induction, maturation, germination, and plant recovery.

The main drawbacks of exogenously applied BA in the micropropagation of horticultural fruit trees include the induction of somaclonal variation, hyperhydricity, shoot-tip necrosis, rooting inhibition, and low *ex-vitro* acclimatization [167], which are physiological disorders that can detrimentally compromise the quality of fruit tree propagules. Compared to BA, the topolin family of CKs has been shown to minimize the effects of in vitro-induced hyperhydricity in some fruit crops such as bananas [109] and apple [102]. However, to the best of our knowledge, there are no reports on the alleviation of somaclonal variation based on CK type. Somaclonal variation can lead to changes in both nuclear and cytoplasmic genomes, which can be genetic or epigenetic [78]. Both BA and *meta*-topolins produced somaclonal variants in *Fragaria* x *ananassa* [97] and *Musa* spp. [109], respectively, for which clonal propagation was the desired outcome. On the other hand, the induction of somaclonal variation may be advantageous and holds great promise in tree cultivar improvement, for example in citrus breeding, due to its recalcitrance to sexual hybridization [78]. Notwithstanding, a thorough understanding of the molecular mechanisms remains critical in the development of fool-proof and efficient mass propagation systems using either organogenesis or somatic embryogenesis models.

Generally, the ‘trial and error’ approach remains the predominant method in the optimization of micropropagation systems of different plant species and sometimes genotypes within a species, which is partly due to the inherent physiological variations in plants. Thus, the main thrust in tissue culture research for the mass propagation of horticultural fruit crops has largely explored and defined the plant-specific threshold limits of different CKs in shoot proliferation and growth (Table 2 and Table 3). In this regard, CKs have been used to achieve several organogenic end points, primarily shoot induction and proliferation, somatic embryo induction, maturation, and subsequent plant recovery. However, as with the first successful establishment of the breakthrough in vitro tomato root culture by White [168], micropropagation requires an equally revolutionary paradigm shift from the predominant ‘trial and error’ approach.

An alternative approach was demonstrated by Werbrouck et al. [165], followed by Bairu et al. [169] and some other related studies [170], where the analysis of endogenous CK metabolites was successfully used to emphasize the differences in BA and *m*T metabolism and correlate these differences with basic micropropagation parameters. It has been repeatedly shown that such an approach can accelerate the entire optimization process of in vitro micropropagation and subsequently even lead to the development of new active PGRs [171]. However, such an approach is still extremely rare in the case of horticultural fruit crops [86].

Although tissue culture as a means of propagation has been extensively studied in several fruit crops (Table 2 and Table 3), it is only recently that the molecular mechanisms associated with shoot organogenesis and development have been compared to processes occurring during natural plant morphogenesis [172]. Research in this area usually investigates how in vitro shoot production occurs from callus, and few studies have compared organogenic molecular pathways where a direct route to shoot regeneration is primary [172]. Processes linked to the molecular controls of CKs in somatic embryogenesis are even more poorly understood as different species, cultivars, and genotypes with varying physiological and genetic backgrounds have been used to study this process [173]. For a long time, callus was referred to as a de-differentiated mass of highly totipotent cells. However, nowadays, there is consensus to accept callus, irrespective of its origin (either derived from root, leaf, or hypocotyl), as resembling a root primordium that is under the forces of auxin-CK cross-talk [172].

This is of particular relevance when a two-step organogenesis protocol is established and where an auxin-supplemented medium is used to initiate callus proliferation prior to shoot induction through the transfer of the callus to a medium rich in CK(s). The transcriptomic profiling of *Arabidopsis* callus tissue revealed the presence of important genes involved in CK signaling, suggesting the intrinsic readiness of the tissue for shoot regeneration even at early stages of callus induction from explants [173]. Although two genes (*CKX5* and *CKX3*) involved in CK degradation were upregulated, no significant changes were observed with other CK signaling-related genes during the incubation. Primordia with the potential for organogenesis are set up at the callus phase, although genes such as *WUSCHEL* (*WUS*) that function in shoot apical meristem patterning and development may not occur at early stages of callus proliferation [174]. Cell fate mediated by a high CK-rich medium is not well-understood, but purine permeates (*PUP*, e.g., *PUP1* and *PUP2*) may be expressed, and these are linked to the transport of CKs [172]. The over-expression of *IPTs* can overcome the need to add CKs to plantlet growth medium, and those mutants that have a loss of function for *IPT* invariably show a significantly decreased capacity for shoot proliferation, but this growth impairment may be overcome when CKs are applied exogenously [175].

### 3.2. General Growth and Health of Fruit Crops/Trees

The architecture and structures of fruit trees influence the resultant crop efficiency and productivity, which is a foremost selection criterion for fruit breeders [176]. Lateral branch development has been reported to be beneficial for increasing the bearing surface and in promoting the early production in horticultural fruits such as apple and the sweet cherry tree [37,177,178,179,180]. Different approaches are often devised to manipulate the structure of a fruit tree for improved performance and productivity [178,179,181]. The application of CKs can improve branching in young trees in nurseries, providing the opportunity to obtain good tree architecture in the future [14,37]. For instance, BA (100 mg/L) effectively stimulated the lateral branching in young apple trees [182].

Similarly, Çağlar and Ilgin [179] observed a significant increase in the total number and total length of laterals per apple tree, following 2–3 applications of BA (200 mg/L). Furthermore, BA treatments also increased the diameter of laterals compared with the control. The effectiveness of combining CKs with other PGRs, especially GA, has been explored by researchers [177,183]. As demonstrated by Zhalnerchyk et al. [180], Neo Arbolin and Neo Arbolin Extra (commercial branching products consisting of BA and GA) stimulated varying degrees of branching in three apple cultivars. Four variants of Arbolin, with differing compositions of BA and GA, effectively promoted laterals in difficult-to-branch apple cultivars [183]. Likewise, various concentrations of Arbolin increased the number of lateral branching in two cultivars of sweet cherry trees [177].

Major abiotic conditions such as drought, extreme temperature, and salinity are factors limiting crop productivity and food security globally [184,185]. Cytokinins, either produced through the alteration of endogenous CK biosynthesis or exogenous application, have been reported as critical phytohormone signals during different abiotic stresses in many plants [22,184,186,187], including apple trees [188]. Some of the adaptive mechanisms induced by *IPT*-transgenic crop plants include improved rooting characteristics (such as increases in total root biomass, root length and root/shoot ratio), which enhance water uptake from drying soils [22,189]. These observations strongly suggest a key role of CKs in controlling root development, differentiation, and architecture in horticultural fruit trees [189]. Although the mechanisms are not that well resolved, Macková et al. [190] reported the overexpression of *CKX1* using the root-specific *WRKY6* promoter that stimulated strong root growth coupled to a concomitant reduction of the stunting effect of the shoot system. This strategy also induced a significant increase in drought tolerance in the overexpressing lines. A major contributing factor to the improved drought tolerance of *WRKY6: CKX1* transgenic plants could be linked to the more extensive root system and higher root-to-shoot ratio. For example, transcriptomic analysis of tomato [51] led to a discovery of several new genes including xanthine/uracil permease family protein and a cytochrome P450 with abscisic acid 8′-hydroxylase activity, whose expression is highly dependent on CK induction that is involved in leaf development. It is now apparent that the CRF genes are implicated in stress responses controlled by CKs and in tomato, organ-specific responses were linked to the strong expression of the *SICRF1* and *SICRF2* genes by cold stress in shoots and oxidative stress in the roots [52]. Thus, there are unique opportunities to exploit such technologies to create plant varieties with novel traits; however, we note the scarcity of information on similar molecular studies in horticultural plants of economic importance.

### 3.3. Preharvest Phase

As applicable with the commonly utilized PGRs in horticulture, frequently CKs are directly applied during the preharvest phase for diverse purposes such as improved morphological structure, facilitation of harvesting, quantitative and qualitative increases in yield, as well as the modification of nutritive chemicals [11,37,191,192]. Particularly, the need to mitigate preharvest fruit drop, promote vegetative growth, enhance flower bud formation, and control the process of fruit ripening is of considerable importance from a commercial perspective [193]. Manipulating fruit yield and quality requires an understanding of the fundamental processes that determine fruit set, maturation, and ripening [15]. Generally, a wide range of desirable characteristics such as nutritional value, flavor, processing qualities, and shelf life determine the overall fruit quality (Figure 2).

#### 3.3.1. Floral and Fruit Development Following Cytokinin Application during Preharvest Phase

There is increasing evidence of the importance of CKs in reproductive biology, which is gained mainly from studies based on model plants, including thale cress (*Arabidopsis thaliana*) and rice (*Oryza sativa*). More often, intracellular CK content is correlated to the increased onset of inflorescence and floral meristem production [194,195], and this supports the idea of applying CKs to enhance fruit production. Flowering, fruit development, and ripening are complex biological processes known to depend on highly coordinated phytohormonal activities and homeostasis in plants [196,197,198]. The desirability of fruit is largely dependent on the final stages of fruit development that involve physiological, biochemical, and, physical–structural changes.

It is now well-established that seed and fruit development are intimately synchronized processes, due to the biosynthesis of different PGRs in seed tissues [7]. Fruit set, which involves the transition of the quiescent ovary into developing fruit, is regulated by cross-talk among the phytohormones and is probably the most critical step in fruit production [197]. Fruit set can occur following successful pollination and fertilization or via parthenocarpy, which produces seedless fruits without fertilization [199,200]. On the other hand, parthenocarpy as a desirable characteristic in fruit production occurs naturally in several important fruit species, notably banana, grape, pineapple, citrus fruits, and apple [7,201]. Phytohormones including CKs, GA, and auxins regulate the natural developmental mechanism of parthenocarpy in different fruits.

Observations in fruit crops such as strawberry (*Fragaria ananassa*), kiwifruit (*Actinidia deliciosa*), raspberry (*Rubus idaeus*), and grape (*Vitis vinifera*) where CK-associated activities or amounts increase internally in immature seeds and developing fruits suggest the significant ontogenic role that CKs play in floral and fruit development [202]. Although the concentration of CKs significantly increases after fertilization and during early stages of fruit development, little is known about their effects during the later stages of fruit ontogeny. Thus, the role of CKs on biochemical and molecular responses during this phase is unclear. Currently, there is no generic scientific evidence to explain the effects of CKs in the ripening phases, as results from exogenous applications of CKs to different species and cultivars lead to inconsistent physiological responses, suggesting that the effects may likely be species or cultivar-specific [202]. Nevertheless, the spatial and temporal specificity of CK biosynthetic genes (particularly *IPTs* and *CKXs*) and those involved in the activation, perception, and signaling associated with intracellular CK-regulated mechanisms are apparent in maturing grapevines [203].

Cytokinins often induce fruit enlargement through cell division and/or cell expansion [14]. The effect of CPPU and *m*T (used at 100 mg/L) in improving the fruit weight of sweet cherry ‘Bing’, which was recorded as a 15% increase, was attributed to enhanced cell division rather than stimulating fruit expansion [204]. Conversely, in kiwifruit (*Actinidia deliciosa*), cell expansion rather than cell division was mainly responsible for the increase in fruit size, as evidenced by a 30% decrease in the number of cells. However, researchers found 89% higher cell diameter in CPPU-treated fruits compared to control fruits [196]. Generally, varying responses in fruit development have been observed when CKs are exogenously applied in a range of fruit crops (Table 4). For example, in pear, 100 mg/L BA substantially improved fruit size without any adverse effect on the yield and fruit shape [205]. In pear, responses to BA application were strictly cultivar-dependent, as the preharvest BA application significantly increased the fruit size of ‘Spadona’ cultivar, while substantial fruit thinning was observed in ‘Coscia’ pear [206]. Apart from the numerous examples with the effects of BA [37], the effectiveness of CPPU, TDZ, and *m*T application (in a limited number of studies) have also been evaluated in many fruit crops such as blueberry, kiwifruit, sweet cherry, grape, and apple (Table 4). In the majority of the examples, the application of CPPU facilitated fruit enlargement. The fruit size and mass were significantly higher in CPPU-treated blueberry (*Vaccinium corymbosum*) when compared to the untreated control [207]. Likewise, the application of BA (100 mg/L) substantially improved the fruit size of ‘Akca’ pear [205].

The combination of CKs with other PGRs such as GA, auxins, and abscisic acid (ABA) have been explored in different fruit crops (Table 4). The most common combination involves BA and GA_(3, 4, 7)_, which is also the constituent of some commercial products, including promalin and cytolin [37]. Fujisawa et al. [207] observed that combining CPPU and GA led to greater fruit size and mass compared to the control or single applications of CPPU and GA. Likewise, a single application of CPPU increased berry size in seedless table grapes (*Vitis vinifera*) ‘Sultanina’, but the most significant effect was observed when CPPU was combined with GA_3_ [208]. These examples above clearly suggest a synergistic influence of the cross-talk among phytohormones.

Nevertheless, the combination of CKs with other PGRs may not confer any additional benefit for floral and fruit development. In apple ‘Fuji’, BA and ABA caused a thinning effect individually, but when combined, ABA provided little or no additional thinning [209]. Seedless table grape cultivars did not show any beneficial or synergistic developmental effects when CPPU was used alone, but when combined with GA, this exposed the undesirable tendency for CPPU to delay fruit maturity [210]. The application of CPPU and GA_4+7_ conferred no additional benefit in the fruit size of the cultivar, apple ‘Fuji’ [211]. The effects of CKs and other phytohormones on fruit set, development, and ripening are not necessarily generic even for cultivars within the same species, illustrating the magnitude of yet unknown intricate mechanisms regulating these processes. Perhaps, there is an association between the level of endogenous CK and fruit development. Seed-bearing avocado (*Persea americana*) fruits were 10-fold larger than seedless fruits under natural growing conditions [212]. The observation was strongly linked to a positive correlation between the rate of fruit growth and the level of endogenous CK in seed tissues [213].

#### 3.3.2. Physical Changes and Yield Responses Following Cytokinin Application during Preharvest Phase

Physical appearance traits such as size and shape are strongly associated with consumer preference, which may strongly influence the economic value of horticultural fruits [61,193,246,247]. As shown in Table 4, different CKs including CPPU, BA, and TDZ have demonstrated their ability to enhance the attributes of resultant fruits in terms of their color, number, and size as well as the length to diameter (L/D) ratio. In most stone fruits, fruit firmness also indicates maturity and quality [247,248]. The flesh firmness of CPPU-treated apple (‘McIntosh’/M.7) had a linear positive response to varying concentrations (1–8 mg/L) of CPPU [227]. In apple ‘Fuji’, Matsumoto et al. [211] demonstrated the effectiveness of 10 mg/L CPPU in the enhancement of flesh firmness when applied 4 days after full bloom (DAFB). Even though TDZ successfully increased the flesh firmness when applied at full bloom (FB), a reduction in flesh firmness was observed when TDZ was applied late, after the FB period in apple [226]. The treatment with 20 mg/L CPPU (applied at marble and pea stages) increased fruit length and diameter as well as fruit weight in mango [231]. Other CKs, including TDZ and BA have also been effective in the L/D ratio in many apple cultivars [209,226]. A positive effect on the L/D ratio was evident when CPPU was combined with other PGRs such as GA_3_ and ABA in table grapes [208,243].

The potential of different CKs such as BA and CPPU on the yield of horticultural fruit crops has been explored by different researchers (Table 4). Stern and Flaishman [206] investigated the effect of BA on the return yield of pear cultivars ‘Spadona’ and ’Coscia’. In ‘Spadona’ fruits, findings indicated that the total increase in yield of large fruit (≥55 mm in diameter) was 88% with 40 mg/L BA (7.5 tonnes/ha versus 4 tonnes/ha in control), while a total increase of 205% in the yield of large fruit (12.2 tonnes/ha) was obtained with 20 mg/L BA. Unlike ‘Spadona’, BA caused significant ‘Coscia’ fruit thinning of about 35% (444 and 765 fruits/tree in BA and control treatments, respectively), which reduced the total yield from 45 to 32 tonnes/ha. As reported by Kulkarni et al. [231], the spray application of mango with CPPU (10 mg/L) at two different stages resulted in the production of 107 kg/tree (10.7 tonnes/ha). This quantity was approximately two-fold higher when compared to the control. In different cultivars of table grapes, the cluster mass increased markedly (64–76%) at varying concentrations of CPPU [240]. Some of the cultivars also demonstrated significantly higher cluster compactness following CPPU treatment. In the study by Williamson and NeSmith [239], differential responses in yield were observed in different cultivars of highbush blueberries when treated with varying concentrations of CPPU. For instance, there was no significant increase in yield for ‘Sharpblue’ and ‘Santa Fe’ (regardless of the concentrations), while 5 mg/L CPPU led to remarkably higher yields of the ‘Star’ cultivar. A similar cultivar-dependent response in yields was observed in two cultivars of blueberries following the application of varying concentrations of CPPU at different time intervals [238]. While CPPU significantly influenced the yield in ‘Climax’ irrespective of the day of application leading to better yields, an early application of CPPU caused a reduced yield production of the ‘Brightwell’ cultivar. When there is no congruence on the influence of CK-related treatments in terms of yield promotion, it is thus important to test their effects on each cultivated variety.

#### 3.3.3. Biochemical and Physiological Changes Following Cytokinin Application during Preharvest Phase

Even though the quality of fruit is often visually defined by the phenotypic characteristics such as fruit size, firmness, and weight, the associated biochemical and physiological changes are fundamentally important during the developmental stages [247]. Several studies have reported a major trade-off between some desirable phenotypic characteristics and fruit quality [196,214,242,243,249]. Fruit ripening is associated with changes in different biochemical and physiological parameters [250], often resulting in the conversion of less appetising green fruits into highly palatable, aromatic, colored, and nutritionally rich fruits [198]. Examples of the biochemical parameters that determine fruit quality include primary metabolites, such as sugars and amino acids as well as secondary metabolites, notably organic acids and anthocyanins, which contribute to fruit flavor. The application of CPPU may lead to a reduction in total soluble solids (TSS) and a concomitant increase in total titratable acidity (TTA), as shown in some fruits such as apple, blueberry, and grapes (Table 5). In mango, the application of varying CPPU concentrations (10 and 20 mg/L) had no significant effects on the TSS, total sugars, reducing sugars, non-reducing sugars, and acidity [231].

The observed variations in patterns of accumulation of sugars and organic acids in parthenocarpic fruits treated with exogenous CKs may be due to differences in cross-talk dynamics between metabolite signaling and phytohormones [249,251]. In addition, different types of soluble sugars can be either upregulated or downregulated in CPPU-treated fruit trees. For example, whereas the concentrations of sucrose, glucose, and fructose decreased during fruit ripening in CPPU-treated kiwifruits, the level of xylose was significantly increased [196]. Such observations illustrate the complex interactions of CKs with other PGRs and signaling molecules that control the growth and development dynamics of horticultural fruit crops.

### 3.4. Postharvest Phase

The postharvest phase is very critical in the production of fruit. Significant postharvest losses are generally incurred at this particular phase due to the perishable nature of fruit crops after harvest. The relevance of CKs during fruit ripening and senescence in planta and as a postharvest application, to perverse quality and extend the shelf life of various fruit crops, are summarized in Table 6 and Table 7.

#### 3.4.1. Effect of Cytokinin Application in Fruit Ripening and Senescence

##### Effect of Cytokinin Application on Fruit Ripening

The biology of fruit ripening is highly complex and tightly regulated with different changes occurring to the ripening fruit, but one of the main changes associated with ripening is linked to colour (Figure 2). In most cases, this involves a loss of green colour and higher accumulation of non-photosynthetic pigments in climacteric and non-climacteric fruits, as influenced by their respiratory activity, which is often associated with ethylene biosynthesis profiles [253,254].

Other changes include firmness (softening by cell wall degrading activities and alterations in cuticle properties), taste (sweetening of the fruit as sugar levels rise and a decline tartness as organic acids lower), and flavor (associated with the production of volatile compounds providing a characteristic aroma) [242,253]. These activities accelerate ripening and senescence processes, which limit the shelf life of fruits [255]. A significant number of studies has demonstrated that exogenous CKs are effective ripening or senescence-inhibition plant regulators [256,257]. Their application can effectively delay senescence and improve the quality of chlorophyll-containing fruits by inhibiting chlorophyll degradation [257,258]. For example, in litchi, the application of 0.1 g/L BA reduced the expression of chlorophyll degradation-related genes and inhibited the activities of chlorophyll degradation enzymes in chlorophyll-containing tissues, therefore extending the shelf life of litchi by 8 days [259]. A study by Itai et al. [260] showed that the preharvest application of CPPU (100 mg/L) retarded ripening due to delayed chlorophyll degradation and low sugar accumulation in persimmons. Although the preharvest application of CPPU (20 mg/L) resulted in higher glucose, fructose, and sucrose contents in kiwi fruit during 6 months of storage, treated fruits also had a higher starch content compared to the control (Table 6). This indicates an inhibition of starch degradation due to the conversion of starch to sugars, thus delaying ripening as supported by higher chlorophyll content [216]. The effect of 10 mg/L CPPU in delaying ripening was also evident in banana [261]. According to the authors, the postharvest application of CPPU suppressed fruit softening by affecting respiration and ethylene production rates. Together with this, the accumulation of soluble reducing sugars and the hue value, and the maximal chlorophyll fluorescence of banana during storage were all retarded. Postharvest application of BA at a varied concentration of 0, 1, 10, and 100 mg/L delayed the de-greening of calamondins in both light and dark conditions and extended shelf life by 9 days compared to the control [254]. The underlying physiological response was the inhibition of ethylene-induced change in fruit color, and the same effect was observed in cucumber [262].

##### The Effect of Cytokinin Application on Fruit Senescence

Senescence is a crucial aspect of the fruit life cycle and directly affects fruit quality and resistance to pathogens [256]. Reactive oxygen species (ROS) such as superoxide anion (O_2_^•−^), hydrogen peroxide (H_2_O_2_), hydroxyl radicals, and singlet oxygen are primary mediators of oxidative damage in plants and are involved in senescence [263]. Moreover, senescent fruits are more susceptible to postharvest decay, fungal pathogens, and diseases, leading to the rapid necrotization of the tissue during storage [252,256,263]. Zhang et al. [264] reported that a postharvest application of BA, at 500 mg/L, inhibited cell membrane deterioration and induced higher defense-related enzyme activities such as superoxide dismutase (SOD) and peroxidase (POD) in peaches. In addition, it also reduced malondialdehyde (MDA) content, which is responsible for lipid peroxidation and oxidative stress and prolonged shelf life by 16 days compared to the control. Similar observations were also reported by Zhang et al. [259], who demonstrated that a postharvest application of BA (0.1 g/L) reduced H_2_O_2_ accumulation, lipid peroxidation, and polyphenol oxidase (PPO) activity in litchi, resulting in an extended shelf life by 8 days compared to the control. The treatment evidently controlled pericarp browning, quality deterioration, decay, and senescence in litchi (Table 7).

#### 3.4.2. Effect of Cytokinin Application in Enhancing Postharvest Fruit Quality Attributes

The decision for subsequent purchases of fruit is strongly dependent upon consumer satisfaction based on texture, flavor, and aroma, which are related to soluble solids content (SSC) (mainly sugars), titratable acidity (TA), volatile and non-volatile phytochemicals [265]. Cytokinins are known to preserve or enhance these quality attributes in some fruits (Figure 2).

##### Effect of Cytokinin Application on Fruit Texture

While a certain degree of softening is desirable in fruit, depending on species and cultivar, excessive softening results in postharvest decay or consumer rejection [270]. The postharvest application of CPPU (10 mg/L) significantly inhibited banana softening by delaying the climacteric peak, thereby delaying ripening and extending the postharvest storage life of the bananas by 6 days compared to controls [261,271]. Similarly, postharvest application of CPPU (10 mg/L) enhanced texture in kiwifruit stored at 0 °C and 95% relative humidity [196]. According to Ainalidou et al. [196], treated fruits had a 2-fold firmness compared to the control, and this extended storage life by 2 months. The authors attributed this to a delayed climacteric response involving respiration and ethylene production rates that were lowered by 2- and 5-fold, respectively, in the treated samples.

Apart from CPPU, BA is also popular for use as a postharvest application, and some of its effects are linked to its capacity to maintain fruit firmness and delayed cell wall degradation and softening in round summer squash and peaches, for example (Table 7). This effect could be attributed to the inhibition of polygalacturonase and pectin methylesterase activities involved in cell wall degradation and softening. There is a possibility of it indirectly promoting the production of cross-link pectic substances in the cell wall that maintain rigidification, thereby increasing the fruit firmness and shelf life by more than 16 days [241,254]. Although the direct mechanisms are still not well understood, further support for this idea is demonstrated in the study of Massolo et al. [257], where an indication of a substantial delay in cell wall dismantling was apparent, when BA (1 mM) was applied in summer squash. In that study, there was prolonged fruit firmness and shelf life for 25 days, and this was due to higher levels of tightly bound polyuronides, recorded at 45%, that led to significantly delayed water loss from the fruits. On the other hand, the controls showed an extreme decrease of firmness and an increase of water-soluble pectin during storage which accelerated cell wall degradation and softening, limiting the shelf life of fruits to 12 days (Table 7).

Although many studies test the effect of exogenous applications on fruits after harvesting, another strategy that has been on trial is the testing of the effectiveness of preharvest applications in quality keeping and postharvest management. It is interesting to note that a preharvest application of 5 mg/L CPPU (relative to untreated controls) significantly increased the shelf life and quality of Thompson seedless grapes by enhancing berry firmness and decreasing the percentage of unmarketable berries when the fruit was stored at ambient temperature for 7 days after harvest [241]. The retention of berry firmness resulting from the CPPU application was suggested to be due to an inhibition of ethylene biosynthesis that prevented a loss in fruit firmness and extending storability by 7 days compared to the control [241,264]. In support of this, preharvest CPPU application significantly reduced weight loss in Thompson seedless grapes [241] and cucumber [266] and the increased storability of fruits by 7 and 10 days, respectively (Table 6; Table 7).

##### Effect of Cytokinin Application on Fruit Soluble Solids Content (SSC) and Titratable Acidity (TA)

The commonly used index values for harvesting and sale of mature fruits are the levels of SSC and TA. Li et al. [55] reported that a preharvest application of CPPU (15 mg/L) effectively inhibited the accumulation of TSS and significantly delayed the degradation of TA in strawberry by about 2-fold compared to the control during storage, which was mainly due to delayed fruit maturity or ripening. Incidentally, there was a 6-day shelf-life extension due to the slow accumulation of SSC and TA in treated fruits. In comparison to treated fruits, the control had significantly higher TSS and lowered TA during storage due to an early attainment of maturity and ripening. Similarly, CPPU (10 mg/L) significantly inhibited the accumulation of sugars (fructose, sucrose, lactulose, tagatofuranose, tallose, and psicofuranose) in kiwifruit by up to 17-fold compared to the control, and it also extended fruit shelf life by 2 months at 0 °C and 95% relative humidity [196]. Sugars, including sucrose, glucose, and fructose, accumulate after starch degradation in fruits [271,272]. As high SSC and ethylene production rates become more pronounced in untreated kiwifruit compared to the treated ones, accelerated ripening and cell wall degradation become important indicators of a shortening of postharvest storage capacity in kiwifruit [196]. Other work that lends support to the application of CPPU as postharvest treatment is linked to the study of Huang et al. [261], who used 10 mg/L CPPU to effectively delay the accumulation of soluble reducing sugars in banana by more than 2-fold, and it also concomitantly extended the shelf life by 16 days due to delayed ripening and softening. By accumulating metabolites such as sugars and organic acids, plant cells produce a lower osmotic potential that generates a turgor pressure resulting in cell expansion, and this process also requires the cell wall to be irreversibly stretched through cell wall loosening [273]. Therefore, the application of CKs is an effective alternative for inhibiting the accumulation of sugars, ripening and prolonging the shelf life of fruits [196,226,261].

##### Effect of Cytokinin Application on Fruit Phytochemicals

The synthesis and accumulation of secondary metabolites in the developing, maturing, and ripening fruits are of critical considerations in terms of quality and acceptance by consumers [24,274]. Li et al. [55] reported that a preharvest or postharvest application of CPPU (10 or 15 mg/L) significantly decreased the total volatile content (esters, alcohols, acids, terpenes, furanones, and others) of strawberry by 65.3% and 87.7% compared to the control before and after storage, respectively, and it extended the fruit shelf life by 6 days. Similar observations were reported by Ainalidou et al. [196], who demonstrated that a preharvest application of CPPU (10 mg/L) resulted in the downregulation of fatty acid hydroperoxide lyase in kiwifruit, cleaving fatty acid hydroperoxides and forming short-chain (C6) aldehydes, which are the volatile primary constituents of the characteristic odor of fruits. According to the authors, this could be responsible for the inhibition of respiration and ethylene production rates, resulting in delayed ripening and an extension of shelf life by 2 months in treated fruits.

The application of CKs has also been reported to enhance phytochemical quality such as phenols, flavonoids, and antioxidant activity [258,275,276], thus maintaining the nutritional quality of fruits. For instance, the postharvest application of BA led to a significant 2-fold rise in phenylalanine ammonia lyase (PAL) activity compared to the control [259]. This enzyme has a key regulatory role in specialized metabolism, as it drives the first committed step of phenylpropanoid synthesis and thus controls the biosynthesis of secondary metabolites, including anthocyanins in plants [276,277]. The influence of BA in treated fruits is correlated to higher contents of anthocyanin, total phenolics, ascorbic acid, and total antioxidant capacity during storage compared to non-treated controls, and even higher levels of chlorophyll are apparent when this PGR is used after harvest (Table 7). Tsantili et al. [269] reported that the postharvest application of CKs induced anthocyanin formation but fortuitously did not raise ethylene concentrations or encourage the softening of olive fruits. Therefore, this demonstrates that the ability of BA treatment to induce PAL activity is favorable for the synthesis and maintenance of anthocyanin content during postharvest storage [259].

##### Effect of Cytokinin Application on Fruit Antioxidant Enzymes

The application of 10 mg/L CPPU significantly enhanced the upregulation of ꞵ-1,3-glucanase and lactoylglutathione lyase defense enzymes in kiwifruit by almost 3-fold compared to the control. This could be possibly responsible for the inhibition of cell wall degradation and softening in treated fruits, as well as an extension of shelf life by 2 months at 0 °C and 95% relative humidity [196]. The ꞵ-1,3-glucanase enzyme catalyzes the endo-type hydrolytic cleavage of ꞵ-1,3-D-glucosidic linkages in ꞵ-1,3-glucans involved in cell wall thickness [278]. In contrast, the lactoylglutathione lyase enzyme is implicated in CK-defense responses, as it primarily functions in the glyoxal pathways generating *S*-lactoylglutathione from toxic methylglyoxal, which increases intracellular levels of ROS [196]. Thus, higher levels of the expression of lactoylglutathione lyase induced by CKs are speculated to be crucial for fruit protection against oxidative stress [279,280], which could be responsible for the inhibition of ripening and senescence in treated fruits [196]. Thus, there is great merit in using CPPU (applied at 10 mg/L), because it controls antioxidant ROS mechanisms and effectively upregulates short-chain type dehydrogenase/reductase-like (SDR) and abscisic acid stress ripening-like proteins (ASR) in kiwifruit. These proteins, which were implicated in ethylene-controlled fruit ripening, were detected at higher levels by Ainalidou et al. (2016) using a gel-based proteome study, where kiwifruit shelf longevity was increased to 2 months in CPPU-treated kiwifruit samples versus controls. A comparative analysis of seven RNA sequence transcriptomes treated with CPPU showed major effects linked to primary metabolism, specifically carbon and amino acid biosynthesis pathways and photosynthesis genes, especially those associated with chlorophyll production and anthocyanin biosynthesis [67]. The application of CPPU most affected and downregulated genes such as PAL, CHS, and F3’H in the pericarp. According to the authors, further investigation is required to reveal the impact of CPPU in litchi fruit maturation at the color break stage.

#### 3.4.3. Effect of Cytokinin Application against Postharvest Physiological Disorders

Physiological disorders such as pericarp browning, chilling injury, bitter pit, shatter, and cracking reduce the quality, commercial acceptability, and shelf life of fruits [244,281]. Pericarp browning is a major postharvest problem for many fruits, which is mainly due to the desiccation of pericarp and degradation of anthocyanin pigments along with the oxidation of phenolic compounds [281,282]. This leads to an excessive accumulation of ROS, which could cause lipid peroxidation, membrane damage, and consequently premature fruit senescence [281,283,284]. Previous studies have demonstrated the great potential of CKs to enhance both antioxidant compounds, such as ascorbate/ascorbic acid (AsA) and glutathione, and enzymes such as SOD, catalase (CAT), ascorbate peroxidase (APX), and peroxidase (POD), which may inhibit fruit quality deterioration during storage [262,282,283]. A study by Zhang et al. [259] showed that the postharvest application of BA (0.1 g/L) effectively reduced ROS and H_2_O_2_ accumulation, PPO activity, and lipid peroxidation in litchi compared to control samples, thereby inhibiting pericarp browning and extending storage by 8 days. This could be attributed to the observed 2-fold rise in SOD, CAT, and APX activities in treated fruits compared to the controls during storage. Reactive oxygen species, for example (O_2_^•−^), are efficiently converted to H_2_O_2_ by the action of SOD, while H_2_O_2_ is destroyed predominantly by APX and CAT [259,262]. Based on Chen and Yang [262] report, a 50 mM BA treatment significantly reduced membrane permeability and lipid peroxidation, as well as delayed the rate of O_2_^•−^ production and H_2_O_2_ accumulation. In that study, the activities of SOD, CAT, APX, and glutathione reductase (GR) in cucumber were positively correlated to the amounts of measured superoxides and extended shelf life under chilling stress conditions throughout a 16-day period, which was reflected by better postharvest longevity within the treated group.

Moreover, adenosine triphosphate (ATP) content was almost 2-fold higher compared to the control, resulting in a higher level of energy charge, thereby inhibiting chilling injury (Table 7). In addition, the fruits exposed to BA had better antioxidant activity and less pericarp browning [259,262]. Evidence of the scavenging of ROS produced within the pericarp [285] and the protection of the fruits from oxidation reactions during storage [282] currently exist. Moreover, preharvest application of CKs is not only beneficial while the fruit is undergoing development: it may have latent effects that are associated with postharvest performance, significantly reducing physiological disorders such as berry shatter and rachis necrosis in grapes during storage [240,241].

In addition to enhanced firmness, reduction in water loss by 4.6% and natural browning by more than 2-fold in CK-treated banana compared to the control during the storage were evident [261,271]. Furthermore, Ainalidou et al. [196] reported that the preharvest application of CPPU effectively reduced weight loss by 33% in kiwifruit compared to the control, and it extended storage by 2 months; thus, CKs are effective in controlling structural damages in fruits. According to Biton et al. [222], postharvest application of BA (40 µg/mL) significantly reduced cuticle cracking by 45% in persimmons. The treatment also reduced naturally occurring black spots by 40% to 50% in treated fruits compared to the control and extended shelf life by 12 weeks (Table 6).

#### 3.4.4. Effect of Cytokinin Application against Fruit Pathogen Infection

Microbial attacks can be severely detrimental to crop yields, causing insurmountable losses if no proper control measures for infections are undertaken. Although the utilization of CK-based approach for controlling plant pathogenic attacks is not popular, the limited studies suggest that CK applications have antibacterial and antifungal activities against different fruit pathogens. Studies highlighting the effectiveness of applied CKs in inhibiting postharvest fungal pathogens in various fruits are presented in Table 6 and Table 7. For example, the preharvest application of ‘Superlon’ (a mixture of GA_4+7_ and BA) at 40 µg/mL resulted in a 45% reduction in the incidence of naturally occurring alternaria black spot (ABS) disease caused by *Alternaria alternata* in persimmon after three months of storage [222]. As demonstrated by Yu et al. [256], BA (20 µg/mL) enhanced the efficacy of *Cryptococcus laurentii*, which is a well-known postharvest yeast antagonist, in reducing postharvest blue mold disease in vivo in apples compared to the controls. Similar observations were reported by Zhang et al. [259], who demonstrated that the antifungal activity of BA (0.1 g/L) effectively inhibited the decay incidence of harvested litchi by more than 2-fold via the inhibition of *Peronophythora litchii* during 8 days of storage.

Mechanisms for protection against microbial diseases need not only be effective once the plant pathogen has established itself, but those strategies that are preventative in the manifestation of disease are also highly desirable therapies in an agricultural setup (Figure 2). Table 7 summarizes some of the studies demonstrating CK-enhanced disease resistance in fruit crops through the elicitation of defense-related enzymes. As an example, the postharvest application of BA alone or in combination with *Cryptococcus laurentii* at the optimal concentration (1000 μg/mL) effectively inhibited mold infection caused by *Penicillium expansum* in pears, which was mainly attributed to the increase in CAT activity, and it extended storage by 6 days compared to the control [252]. Zhang et al. [264] recorded higher SOD, POD, and PPO activities in peach wounds inoculated with 500 mg/L BA, successfully controlling the brown rot caused by *Monilinia fructicola* with a 63% reduction in treated fruits compared to those that were untreated. As shown by Zhang et al. [259], BA (0.1 g/L) significantly inhibited the decay incidence of harvested litchi via the inhibition of *Peronophythora litchii* during 8 days of storage, which was attributed to the increase of PAL activity, the key enzyme involved in the biosynthesis of phenolic compounds and the upregulation of defense signals in the fruits. As BA appears to control microbial plant infections, it may be prudent to explore other CKs or compounds with CK-like activity in the future. Thus, the priming of fruit crops to impart defense against plant pathogens is gaining popularity as a new technological application.

## 4. Concluding Remarks and Future Prospects

In general, the intrinsic complexity of the function of CKs in plants requires a highly integrated and interdisciplinary approach that spans across different fields of plant sciences, for the artistic exploitation of these important regulators of plant growth and development in horticultural fruit crops. Although the use of CKs is a routine practice in plant propagation, mass propagation, and agronomic field setups, their application in postharvest fruit management is limited. New opportunities for their exploitation in postharvest fruit processing are evident.

The high variability in response to CK applications is highly dependent on multiple factors that require considerations in practice. Synthetic compounds such as CPPU, a phenylurea synthetic CK that has become one of the widely used chemicals with CK-like effects in fruit production globally, still show inconsistencies in its effects, which may depend on factors such as genotype (cultivar), phenological stage, concentration, and synergistic interactions arising from combinations with other PGRs [14]. The CK type, whether it be natural or synthetic, also leads to variable effects at different phases of fruit crop production. For fruit development, CPPU, TDZ, and *m*T appear to be the most effective CKs, as demonstrated in some fruit crops, but these effects are never necessarily all inclusive. Although not necessarily covered in this particular review, microbial communities that form associations with plants may contribute to endogenous CK pool. At present, the relationship between microbiome-synthesized CKs and growth-promoting effects is still not well understood or resolved. However, despite their routine applications in many different species, knowledge gaps about influences of CKs in fruit development still exist. Mechanisms of action linked to exogenous CKs in various fruit crops (except for model plants) are barely understood. This is largely reflective of a past mindset in applied agricultural research that did not necessarily examine questions related to the mechanistic aspects; thus, the fundamental information of these processes remains unknown. Next-generation technologies have provided insights into global transcriptome changes in relation to CK-responsive genes, and many of these are involved in signaling, metabolism, and transport mechanisms that control plant growth and development [51]. Genome-wide studies at the transcriptome and proteome levels may reveal interactive protein–protein networks that ultimately contribute to regulating biological growth processes in developing fruits. Studies that provide global insights into molecular mechanisms and genes involved in both preharvest and postharvest biological processes that may be appropriate targets in the development of novel strategies for crop improvements are starting to emerge. However, not all details are available in terms of the basic growth and developmental effects of CKs in fruit crops, as there are unresolved issues related to shoot branching, root branching, and fruit production. The development of new innovative technologies that will exploit the different biological actions of CKs in plants requires an understanding of the modes of action, target sites, genetic-to-metabolomic networks, CK-responsive target genes, and their functional network activity, responses to environmental stresses, and climate changes, to name a few. Such fundamental knowledge is critical for the exploitation of CKs on a broader range of crops, especially for the postharvest quality preservation of fruits. Many investigations that test the effect of CKs on stress are under controlled environments, indicating that CKs are modulators of stress acting through a growth-defense trade-off [53]. In natural and/in-field agricultural settings, influences of CKs on genotype–environment interactions are still poorly understood, and future studies should focus more rigorously on testing CK stress modulation in natural environments for agricultural ecosystem management, especially in the context of climate change. The application of CK as priming agents to engage plant immunity for biotic stress responses and biotrophic is evident in the literature. However, the adoption of priming technology to circumvent or prepare plants for coping with abiotic stresses is rare, and this is especially true for horticultural fruit crops.

While fruit expansion is a key event, there is limited literature covering the role of CKs in the transition from the cell division to the expansion phases and the sustained growth of the fruit. Besides, remarkably few authors have examined if other phytohormones contribute to regulating the production of volatile chemicals during ripening. With other ripening characteristics completely overlooked, it is becoming clear that without a complete understanding of the role of CKs in fruit improvement and management, novel biotechnologies, notably CRISPR/Cas9 mutagenesis, may be more challenging to implement. Despite the proven importance of CKs for micropropagation, Karkute et al. [26] emphasized that the lack of reproducible in vitro regeneration protocols for many fruit crops is hindering the wide application of CRISPR/Cas9 technologies. For many fruit crops, artificial mutant technology is not easy to exploit, as reference genome sequence information is not always available. This information could assist with our understanding of genetic mechanisms that underpin the functioning of CKs. Therefore, such advances will require the use of a broader range of horticultural fruit crops as new models to provide a better understanding of the broader functioning of CKs and their regulatory controls in developmental pathways. This understanding is important in order to increase output and meet the goals of food security and sustainable fruit production.

## Figures and Tables

**Figure 1 biomolecules-10-01222-f001:**
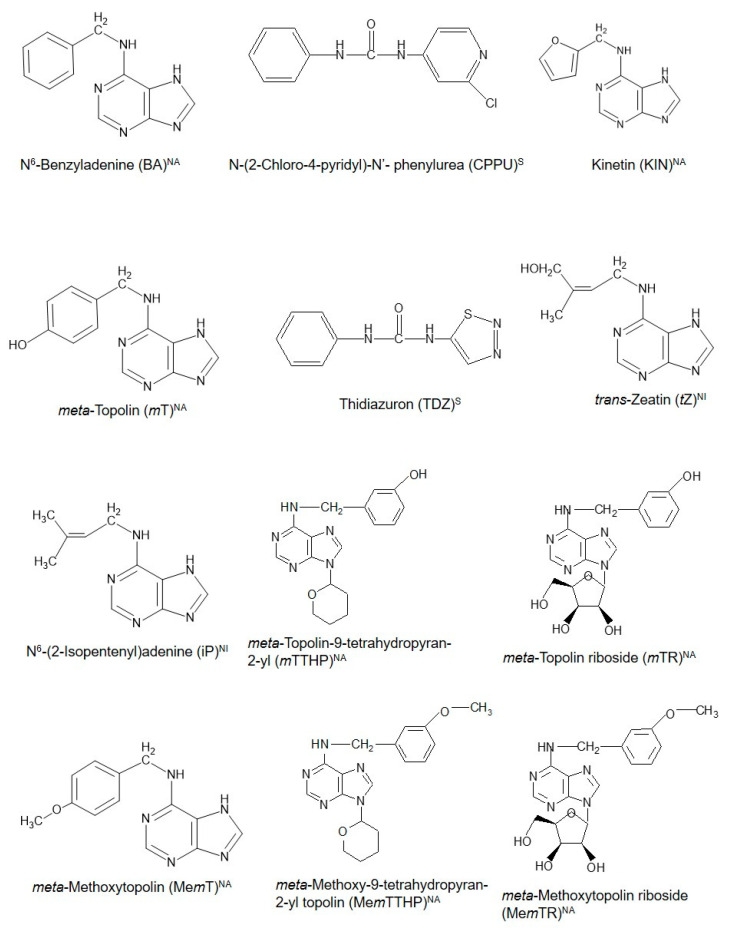
Chemical structures of cytokinins (CKs) used in propagation, preharvest, and postharvest stages during the production of some common horticultural fruit crops. NA = natural aromatic CK; NI = natural isoprenoid CK, and S = synthetic CK.

**Figure 2 biomolecules-10-01222-f002:**
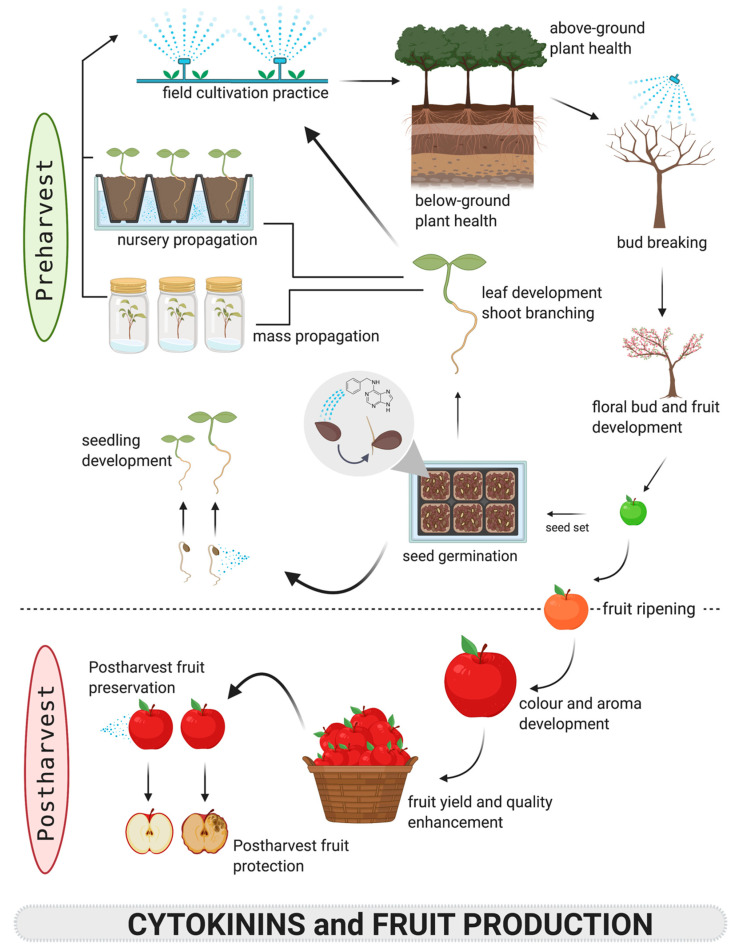
An overview of cytokinin applications in the production of horticultural fruits using apple as an example.

**Table 1 biomolecules-10-01222-t001:** Gene expression-related responses to cytokinin application in different horticultural fruit crops.

Attribute and Fruit	Focus of Study	Response(s)	Reference
Seed and Flowering			
San Pedro fig ‘Asteran’ *Ficus carica* L.	Effect of CPPU on the parthenocarpy induction	CPPU upregulated phytohormone genes such as *GA20ox*, *GA3ox*, *GID1*, *GID2*, *AUX/IAA,* and *GH3*, while downregulating *NCED*, *PP2C* and *ABF*	[63]
Strawberry *Fragaria vesca* L.	Effect of IAA, BA, ACC, and GA_3_ on *FvPHL* gene regulation during seedling development	BA, ACC and GA_3_ slightly regulated the gene expression of *FvPHL3/5/6* by BA, *FvPHL5* by ACC and GA3, and *FvPHL3* by IAA, while ABA influenced the expression of all six *FvPHL* genes	[64]
Apple “Fuji’ *Malus domestica* Borkh.	Effect of BA (5 mM), decapitation, and lovastatin on the expression of *MdIPT* and *MdCKX* genes in apple during axillary bud outgrowth	BA and decapitation treatment induced the upregulation of *MdIPT*, *MdCKX,* and *MdPIN1* genes, while lovastatin (a compound that effectively suppresses axillary bud outgrowth) inhibited gene expression. Both BA and lovastatin upregulated *MdCKX8* and *MdCKX10* genes	[57]
Apple *Malus domestica* Borkh.	Effect of 300 mg/L BA on floral genes (*MdFT*, *AFL1,* and *MdTFL1*) during flower development after 180 DAF	BA upregulated the transcription of *MdFT* at 110 DAF, *AFL1* at 50 and 110 DAF, with a significant decline in *MdTFL1* expression at 30 and 180 DAF	[58]
Fruit quality			
Apple “Pink Lady’ *Malus* x *domestica* Borkh.	Effect of GA_4+7_ and BA on the cellular mechanism of calyx-end cracking during fruit development	Early application of GA_4+7_ and BA (commercial product Superlon™ with 1.9% (*v*/*v*) of both plant growth hormone) increased epidermal cell density, which strengthened cell-wall components and upregulated the expression of genes responsible for fruit-cracking resistance	[65]
Biochemical and physiological parameters			
Kiwi ‘Hayward’ *Actinidia chinensis* var. *deliciosa*	Effect of CPPU on transcript abundance of carbohydrate metabolism genes at a standard and a high carbohydrate supply	CPPU-treated fruits reduced starch synthesis while increasing starch degradation during early fruit development (standard carbohydrate supply). However, CPPU-treated fruits increased vacuolar invertase transcripts, which in turn increased the sucrose cleavage associated with increased fructokinase (*FK4*) gene expression in early fruit development (high starch supply)	[56]
Kiwi ‘Xuxiang’ *Actinidia deliciosa.*	Effect of CPPU on volatile emissions and differential gene expression related to these compounds after days of storage	CPPU inhibited the biosynthesis of volatile compounds including aldehydes, esters, and terpenes. CPPU influenced gene expression related to hormone signal transduction in aldehydes, alcohols, and terpene biosynthetic pathways	[66]
Strawberry ‘Akihime’ *Fragaria × ananassa* Duch.	Effect of CPPU application on the proteomic analysis during pre and postharvest	In total, 88 and 56 proteins were expressed during harvest and after storage, respectively. CPPU regulated glycolysis, photosynthesis, and acid metabolism before storage. Particularly, the upregulated expression of the *LDOX* gene contributed to the induction of anthocyanin content in strawberry in response to CPPU. However, CPPU suppressed volatile biosynthesis	[55]
Litchi ‘Feizixiao’ *Litchi chinensis* Sonn.	Effect of 25 mg/L ABA and 4 mg/L CPPU on physiological changes and transcriptome profiling	ABA upregulated the expressions of genes (*LcGST4*) involved in flavonoid and anthocyanin biosynthesis, while CPPU induced genes related to carbon metabolism, amino acids, photosynthesis, and downregulated genes related to anthocyanin biosynthesis	[67]
Litchi ‘Feizixiao’ *Litchi chinensis* Sonn.	Effect of ABA and CPPU on the expression of anthocyanin-related *LcGST4* genes	ABA enhanced anthocyanin accumulation through the induced expression of *LcGST4* during ripening stages. CPPU reduced anthocyanin production and *LcGST4* expression remained at low levels	[68]
Pear ‘Cuiguan’ *Pyrus pyrifolia* Nakai	Effect of CPPU in verifying the function of *B-PpRR*s during fruit coloration and anthocyanin production in pear that never produce anthocyanin	CPPU stimulated anthocyanin production in the skin of fruitlets after 16 days of treatment. CPPU also induced *B-PpRR* anthocyanin biosynthetic genes, which are presumed to mediate anthocyanin production	[54]
Grape ‘Neo Muscat’ *Vitis vinifera* L.	Mutagenesis: expression of *Vitis vinifera phytoene desaturase (VvPDS)* gene	Carotenoid biosynthesis	[69]
Sweet orange ‘Valencia’ *Citrus sinensis*	Mutagenesis: expression of *Citrus sinensis Phytoene desaturase (CsPDS)* gene	Carotenoid biosynthesis	[70]
Abiotic and biotic effects			
Wanjincheng orange *Citrus sinensis* Osbeck	Mutagenesis: expression of *Citrus sinensis Lateral organ boundaries 1 (CsLOB1)* promoter	Citrus canker resistance	[71]
Duncan grapefruit *Citrus paradisi*	Mutagenesis: expression of *CsLOB1*	Citrus canker resistance	[72]
Strawberry ‘Akihime’ *Fragaria × ananassa* Duch.	Effect of CPPU application on the proteomic analysis during pre- and postharvest	In total, 88 and 56 proteins were expressed during harvest and after storage, respectively. CPPU application resulted in higher capacity of resistance in strawberry to stress stimuli after storage	[55]
Melon ‘Yangjiaomi’ *Cucumis melo* L.	Effect of *t*Z application on *TCS* genes	Type-A RRs, *CmRRA1* - *CmRRA7*, were upregulated after 2 h of *t*Z application	[73]

ABA = abscisic acid; ACC = 1-aminocyclopropane-1-carboxylic acid; BA = *N*^6^-benzyladenine; CPPU = *N*-(2-chloro-4-pyridyl)-*N*´-phenylurea; DAF = days after flowering; GA = gibberellins; IAA = indoleacetic acid; *LOB1* = *Lateral organ boundaries 1;* PDS = *phytoene desaturase*; *PHL* = *PH-LIKE*; RRs = response regulators; TDZ = thidiazuron; *t*Z = *trans*-zeatin; TCS = two-component system.

**Table 2 biomolecules-10-01222-t002:** Cytokinin applications for in vitro propagation protocols of horticultural fruit crops.

Species	Cultivar/Accession	Explant	Factor(s) Investigated	Major Outcome(s)	Reference
Hardy kiwifruit *Actinidia arguta* Planch.	Lv Wang, Kui Lv	Anthers	Effect of different IAA, 2,4-D, BA, and KIN concentrations on callus induction	Callus induction was best achieved with 2,4-D and BA	[84]
Saskatoon berry *Amelanchier alnifolia* Nutt.	Northline	Shoot-tips	Effect of BA (0, 8.88, 13.3, 17.7, and 22.2 µM) on shoot proliferation	22.2 µM BA yielded a maximum number of shoots (12)	[85]
Saskatoon berry *Amelanchier alnifolia* Nutt.	Pembina	Shoot-tips	Effect of BA (0, 8.88, 13.3, 17.7, and 22.2 µM) on shoot proliferation	The highest number of shoots (13) were produced with 17.7 µM BA treatment	[85]
Saskatoon berry *Amelanchier alnifolia* Nutt.	Smoky	Shoot-tips	Effect of BA (0, 8.88, 13.3, 17.7, and 22.2 µM) on shoot proliferation	Maximum shoots (15) produced in 22.2 µM BA treatment	[85]
Saskatoon berry *Amelanchier alnifolia* Nutt.	Thiessen	Shoot-tips	Effect of BA (0, 8.88, 13.3, 17.7, and 22.2 µM) on shoot proliferation	BA (22.2 µM) produced a high number of shoots (21)	[85]
Saskatoon berry *Amelanchier alnifolia* Nutt.	Prince Williams	Shoot-tips	Effect of different CKs (iP, BA, *m*T, and *m*TTHP) on organogenesis	20 and 10 μM *m*T improved shoot proliferation and regenerant quality, respectively. 1 μM *m*TTHP treatment had the highest root proliferation (3 roots/explant)	[86]
Pineapple *Ananas comosus* (L.) Merr		Shoot-tips	Effect of BA (0, 0.2, 0.4, 0.6, 0.8, and 1.0 mg/L) on shoot proliferation	Maximum shoots (9) were achieved with 1 mg/L BA during a culture interval of 8 weeks	[87]
Pineapple *Ananas comosus* (L.) Merr.	Pattawia	Shoot-tips	Effect of *m*T (0, 2.5, and 5 µM) on shoot proliferation	2.5 µM *m*T (shaking liquid culture) had the highest number of shoots (16 shoots)	[88]
Jackfruit *Artocarpus heterophyllus* Lam.		Seedlings	Effect of BA (0, 3.0 mg/L) and TDZ (0, 0.5, 1.0, and 1.5 mg/L) on shoot induction and proliferation	BA (3.0 mg/L) resulted in the high shoot induction, multiple shoot formation, and shoot size	[89]
Chokeberry *Aronia melanocarpa*	PI 613016	Cotyledons	Effect of NAA concentration in combination with 10 μM BA on shoot organogenesis	Combination of 1 μM NAA + 10 μM BA improved shoot number (3)	[90]
Chokeberry *Aronia mitschurinii*	Viking	Leaf	Effect of different concentrations of NAA, IBA, 2,4-D, BA, and TDZ on shoot regeneration	5 μM IBA + 10 μM TDZ had the highest number of shoots	[90]
Chokeberry *Aronia mitschurinii ×Sorbaronia fallax*		Cotyledons	Effect of NAA concentration in combination with 10 μM BA on shoot organogenesis	Maximum number of shoots (4) were produced from 5 μM NAA + 10 μM BA	[90]
Sour orange *Citrus aurantium* L.		Epicotyl segments	Effect of BA, iP, TDZ, KIN, and CPPU on shoot organogenesis	Shoot initiation (70%) and number of shoots (2) were achieved in 0.05 mg/L CPPU treatment	[91]
Orange *Citrus reticulate* x *Citrus Poncirus trifoliate*	‘Sunki’ x ‘Benecke’	Epicotyl segments	Effect of CKs (BA and *m*T) alone or in combination on shoot proliferation	BA (1 µM) produced 20 shoots	[92]
Kinnow mandarin *Citrus reticulata* L.		Immature fruits	Effect of 2,4-D, BA, and KIN on plantlet regeneration via somatic embryogenesis	1 mg/L BA induced somatic embryos, while 5 mg/L 2,4-D in combination with 1 mg/L BA enhanced maturation of somatic embryos	[93]
Citrus rootstocks *Citrus volkameriana*		Node	Effect of BA (0, 0.5, 1, 2, and 4 mg/L) on shoot proliferation	1 mg/L BA had the highest number of shoots and leaf explants while 0.5 mg/L improved shoot length	[94]
Fig *Ficus carica*	Japanese BTM 6	Axillary shoot tips	Effect of BA and Z (0, 0.5, 1.0, 1.5, and 2.0 mg/L) on shoot multiplication	2 mg/L BA produced the highest number of shoots (1.67 ± 0.33) that were relatively long (0.51 ± 0.07 cm)	[95]
Strawberry *Fragaria x ananassa*	Calypso	Apical buds	Effect of PGRs (TDZ, BA, 2,4-D, and IBA) on shoot proliferation	0.5 mg/L TDZ in combination with 0.02 mg/L 2,4-D yielded 12 shoots	[96]
Strawberry *Fragaria x ananassa*	Sveva	Apical buds	Effect of PGRs (TDZ, BA, 2,4-D, and IBA) on shoot proliferation	Maximum shoots (10) were produced in 3 mg/L BA (with 0.2 mg/L IBA) treatment	[96]
Strawberry *Fragaria x ananassa* Duch.		Runner tips, shoots, leaves, nodal segments	Effect of different concentrations of BA on inducing somaclonal variants	High concentration (6 mg/L) of BA induced somaclonal variation	[97]
Strawberry *Fragaria x ananassa* Duch.	Santa	Runner tips	Effect of KIN (0, 0.5, 2.0, 3.0, and 4.0 mg/L) on shoot regeneration	Shoot induction was achieved at 0.5 mg/L concentration	[98]
Strawberry *Fragaria x ananassa* Duch.	Fanta	Runner tips	Effect of KIN (0, 0.5, 2.0, 3.0, and 4.0 mg/L) on shoot regeneration	Shoot induction was achieved at 0.5 mg/L concentration	[98]
Strawberry *Fragaria x ananassa* Duch.	Berrystar	Runner tips	Effect of KIN (0, 0.5, 2.0, 3.0, and 4.0 mg/L) on shoot regeneration	Shoot induction was achieved at 0.5 mg/L concentration	[98]
Strawberry *Fragaria x ananassa* Duch.	Honeybell	Runner tips	Effect of KIN (0, 0.5, 2.0, 3.0, and 4.0 mg/L) on shoot regeneration	Shoot induction was achieved at 0.5 mg/L concentration	[98]
Strawberry *Fragaria x ananassa* Duch.	Okhyang	Runner tips	Effect of KIN (0, 0.5, 2.0, 3.0, and 4.0 mg/L) on shoot regeneration	Shoot induction was achieved at 0.5 mg/L concentration	[98]
Litchi *Litchi chinensis* sonn.		Zygotic embryos	Effect of different BA concentrations (0.004, 0.02, 0.04, 0.2, and 0.4 μM) on germination and plantlet regeneration from encapsulated somatic embryos	Low BA concentration (0.004 μM) increased seed germination and plantlet development	[99]
Apple *Malus domestica* Borkh.	M.26	Young leaves	Effect of different CKs on shoot proliferation	18.20 µM BAR increased shoot number (3)	[100]
Apple *Malus domestica* Borkh.	Red Fuji	Leaves	Post-effects of PGR levels of proliferation media on rooting of in vitro shoots	95% rooting in shoots was achieved with *m*T (20.7 µM) and BA + KIN (4.4 + 7.0 µM) with IBA	[101]
Apple *Malus domestica* Borkh.	Royal Gala	Leaves	Effect of pre-conditioning with BA (0.5 mg/L) and *m*T (0.5, 1, 1.5, and 2 mg/L) on the morphogenic activity of regenerants	*m*T (0.5 and 1.5 mg/L) significantly decreased hyperhydricity (down to 13.4%) and increased the number of shoots per leaf segment (up to 15.1)	[102]
Apple *Malus domestica* Borkh.	Royal Gala	Shoots	Effect of CKs on structural characteristics of leaves and their post-effects on subsequent shoot regeneration	Maximum shoots were produced in 2.1 µM *m*T treatment	[103]
Apple *Malus domestica* Borkh.	Royal Gala	Leaves	Effect of different CKs on shoot proliferation	2.27 µM TDZ resulted in the production of 11 shoots per explant	[100]
Apple *Malus domestica* Borkh.	Royal Gala	Leaves	Effect of TDZ (0.5 mg/L), BA (5 mg/L), BAR (5 mg/L), and *m*TR (6.5 mg/L) during regeneration on in vitro rooting	No rooted shoots were obtained after shoot induction with TDZ and *m*TR while 10% and 25% of shoots developed roots with BA and BAR, respectively	[104]
Apple *Malus domestica* Borkh.	Cacharela, Camoesa, Repinaldo, Tres en Cunca, Gravillán, Ollo Mouro, José Antonio, Príncipe Grande	Apical buds	Shoot multiplication of eight different apple cultivars using four CKs: BA, Z, iP, and TDZ at varying concentrations (0, 0.25, 0.5, and 1 mg/L)	0.5 and 1.0 mg/L BA improved shoot multiplication in apple cultivars	[105]
Mulberry *Morus alba* L.	Variety S-1	Nodal explants	Effect of different concentrations of BA, KIN, and iP on shoot regeneration	Maximum shoot regeneration was achieved from 0.5 mg/L of BA	[106]
Banana *Musa acuminata* Colla	Grand Naine	Shoot apex	Effect of BA (0, 10, 20, and 30 μM) in combination with NAA (1.0 μM) on shoot proliferation	8.26 shoots per explant were generated from 20 μM BA, with 10 μM BA producing 6.18 shoots per explant and 30 μM BA having 7.94 shoots per explant	[107]
Banana *Musa* spp.	Bwara (AAA-EA)	Shoot-tips	Effect of different concentrations (16.8, 20.8, 24.8, and 28.8 µM) of BA, TDZ, Z, iP, and KIN on shoot proliferation	28.8 µM BA increased shoot number (8)	[108]
Banana *Musa* spp.	Grand Naine	Shoot-tips	Effect of CKs (*m*T, *m*TR, Me*m*T, Me*m*TR, and BA) on shoot proliferation and somaclonal variation	Highest shoot proliferation (20 shoots) was established with 15 µM *m*T and *m*TR. All the tested cytokinins did not prevent somaclonal variation	[109]
Banana *Musa* spp.	Kibuzi (AAA-EA)	Shoot-tips	Effect of different concentrations of BA, TDZ, Z, iP, and KIN on shoot proliferation	Highest number of shoots (6) occurred in 5.68 µM TDZ	[108]
Banana *Musa* spp.	Ndiziwemiti (ABB)	Shoot-tips	Effect of different concentrations (16.8, 20.8, 24.8, and 28.8 µM) of BA, TDZ, Z, iP, and KIN on shoot proliferation	Efficient shoot proliferation (9 shoots) was achieved with 6.81 µM TDZ	[108]
Banana *Musa* spp.	Williams	Shoot-tips	Effect of CKs (BA and topolins) on shoot proliferation	Maximum shoots (7) were produced with 30 µM *m*T	[110]
Banana *Musa* spp.	Williams	Shoot-tips	Effect of CKs (*m*T, *m*TR, Me*m*T, Me*m*TR, and BA) on shoot proliferation and somaclonal variation	Highest proliferation (20 shoots) was achieved with 30 µM *m*T and *m*TR. All the tested cytokinins did not prevent somaclonal variation	[109]
Banana *Musa* spp.	Zelig	Shoot-tip	Effect of CKs (BA and TDZ) on somaclonal variation	2.5 mg/L BA had the least level (40%) of somaclonal variants	[111]
Plantain *Musa* spp.	CEMSA 3⁄4	Shoot-tips	Effect of different concentrations (1.33, 2.22, 4.44, 13.3, and 22.2 µM) of BA and *m*T on shoot proliferation	*m*T (4.4 µM) produced the maximum number of shoots	[112]
Plantain *Musa* spp.	CEMSA 3⁄4	Shoot-tips	Effect of different concentrations (0, 1.3, 2.2, 4.4, 13.3, and 22.2 µM) of BA, *m*T, and TDZ on shoot proliferation	Shoot number (11) was highest with 4.4 µM *m*T treatment	[113]
Wild Amazonian passion fruit *Passiflora cristalina*		Seeds	Effect of different concentrations of BA, TDZ, and KIN on the induction of *de novo* organogenesis	BA improved shoot organogenesis	[114]
Passion fruit *Passiflora edulis* Sims	FB-300 Araguary	Seeds	Effect of different concentrations of BA, TDZ, and KIN on shoot organogenesis	TDZ was the only type of CK that induced shoot production	[115]
Avocado *Persea americana* Mill.	Hass (scion) and a Mexican seedling (IV-8) as rootstock	Nodal segments	Effects of three concentrations of BA on shoot proliferation	BA (1.3 µM) treatment produced longer shoots with more leaves	[116]
Citrus rootstocks *Poncirus trifoliata*	Flying Dragon	Node	Effect of BA (0, 0.5, 1, 2, and 4 mg/L) on shoot proliferation	1 mg/L BA had the highest number of shoots, explant leaves and shoot length	[94]
Citrus rootstocks *Poncirus trifoliata*	Serra	Node	Effect of BA (0, 0.5, 1, 2, and 4 mg/L) on shoot proliferation	2 mg/L BA produced maximum number of shoots, explant leaves and shoot length	[94]
Citrus rootstocks *Poncirus trifoliata*	Rubidoux	Node	Effect of BA (0, 0.5, 1, 2, and 4 mg/L) on shoot proliferation	Low BA (0.5 mg/L) concentration yielded a number of shoots and leaf explants with the control treatment producing long shoots	[94]
Citrus rootstocks *Poncirus trifoliate* x *Citrus paradisi*	*Citrumelo* ‘Swingle’,	Node	Effect of BA (0, 0.5, 1, 2, and 4 mg/L) on shoot proliferation	Highest number of shoots and leaf explants were produced with 1 mg/L BA whereas 4 mg/L BA increased shoot length	[94]
Citrus rootstocks *Poncirus trifoliate* x *Citrus sinensis*	*Citrange* ‘Carrizo’	Node	Effect of BA (0, 0.5, 1, 2, and 4 mg/L) on shoot proliferation	2 mg/L BA produced the highest number of shoots and explant leaves while 1 mg/L BA improved shoot length	[94]
Pomegranate *Punica granatum* L.	Kandhari Kabuli	Mature leaves	Effect of BA (0.5–1.5 mg/L), KIN (0.10–0.50 mg/L) and NAA (0.25–0.50 mg/L) on callus induction and shoot regeneration	Maximum callus induction was obtained from a combination of 4 mg/L NAA and 2 KIN. 1.5 mg/L BA with 0.5 mg/L NAA and 0.25 mg/L KIN improved shoot induction, shoot number, and shoot length	[117]
Pear *Pyrus communis*	Barburiña, Manteca Oscura	Apical buds	Effect of BA, Z, iP, and TDZ at 0, 0.25, 0.5, and 1 mg/L on shoot proliferation	Shoot multiplication was highest at 0.5 and 1.0 mg/L BA treatment in both cultivars	[105]
Pear *Pyrus communis*	Bartlett	Cotyledons	Effect of NAA concentration in combination with 10 μM BA on shoot organogenesis	NAA (1 μM) in combination with BA (10 μM) improved shoot organogenesis	[90]
Sweet cherry *Prunus avium* L.	Lapins	Lateral buds	Effect of BA, KIN, iP, and TDZ (1, 2, 5, 10, and 15 μM) in combination with IBA (0, 0.5, 2.5, and 5 μM) on shoot multiplication	BA produced the highest number of shoots with iP, TDZ, and KIN having poor multiplication. However, KIN and iP yielded sturdy shoots	[118]
*Prunus microcarpa* subsp. *tortusa*		Cotyledonshypocotyls, roots	Effect of BA, *m*T, and TDZ on shoot regeneration	BA and *m*T had the higher shoot regeneration relative to TDZ	[119]
Cherry rootstock *Prunus fruticosa* × *Prunus lannesiana*	Krymsk^®^ 5 (cv. VSL 2)	Nodal segments	Effect of six CKs: four adenine type (BA, iP, KIN, and *m*T) and two phenylureas (TDZ and CPPU) at different concentrations (2.4, 4.8, and 9.6 μΜ) on shoot multiplication	Maximum shoot number (3.5 shoots at 9.6 μΜ) and node per explant (10 nodes at 9.6 μΜ) number were obtained from BA, while *m*T produced the highest number of nodes per cm and nodes per shoot	[120]
Mountain ash *Sorbus aucuparia ×Sorbaronia dippelii*		Cotyledons	Effect of NAA concentration in combination with 10 μM 6-BA on shoot organogenesis	Combination of 5 μM NAA with 10 μM BA improved shoot organogenesis	[90]
Mountain ash *Sorbus aria*		Cotyledons	Effect of NAA concentration in combination with 10 μM BA on shoot organogenesis	BA (10 μM) and NAA (5 μM) improved shoot organogenesis	[90]
Blueberries *Vaccinium corymbosum* L.	Sunshine Blue	Leaf, stem, and callus	Effect of Z and IBA on adventitious shoot regeneration	Z (9.12 µM) and IBA (2.46 µM) improved shoot regeneration in leaf and callus explants, while Z (9.12 µM) and IBA (1.23 µM) showed maximum shoot number in stem explants	[121]
Blueberries *Vaccinium corymbosum* L.	Bluejay	Leaf	Effect of Z and IBA on adventitious shoot regeneration	Callus formation and shoot number improved with Z (9.12 µM) and IBA (2.46 µM)	[121]
Blueberries *Vaccinium corymbosum* L.	Top Hat	Leaf	Effect of Z and IBA on adventitious shoot regeneration	IBA (1.23 µM) and Z (9.12 µM) increased callus formation and shoot number	[121]
Blueberries *Vaccinium virgatum*	Pink Lemonade	Leaf	Effect of Z and IBA on adventitious shoot regeneration	Control yielded high number of shoots, while Z (9.12 µM) and IBA (4.92 µM) improved callus formation	[121]
Blueberries *Vaccinium corymbosum* L.	Bluejay	Young stems	Effect of BA (2.22, 4.44, and 6.66 μM) and Z (2.28, 4.56, and 6.84 μM) with 0.57 μΜ IAA on shoot proliferation	6.84 μM Ζ increased shoot number per explant and 2.28 μM Ζ produced longer shoots	[122]
Blueberries *Vaccinium ashei* Reade rabbiteye hybrid derivative	Pink Lemonade	Young stems	Effect of BA (2.22, 4.44, and 6.66 μM) and Z (2.28, 4.56, and 6.84 μM) with 0.57 μΜ IAA on shoot proliferation	The highest number of shoots were obtained from 6.84 μM Ζ, with 2.28 μM Ζ treatment producing longer shoots	[122]
Lowbush blueberry *Vaccinium angustifolium* Ait.	Fundy and two wild clones (‘NB1′ and ‘QB1′)	Shoot-tips	Performance of ‘Fundy’, ‘QB1′, and ‘NB1′ blueberries on shoot proliferation in liquid bioreactor cultures with 1 µM Z over two subculture	Genotypes differed significantly with respect to proliferation with ‘NB1′ producing 8.5 shoots/explant after 8 weeks	[123]
Highbush blueberry *Vaccinium corybosum* L.	Berkeley	Shoots	Effect of 0.5 mg/L Z with IBA (0.1, 1 and 5 mg/L) on shoot organogenesis	Maximum shoots (2) produced in 0.5 mg/L Z (with 0.1 mg/L IBA) treatment	[124]
Highbush blueberry *Vaccinium corybosum* L.	Biloxi	Nodal segments	Effect of different CKs (TDZ, Z, and ZR) alone or in combination with NAA (2.69 µM) on shoot organogenesis	Plant regeneration rate (88%) and shoot number (6) were achieved with 11.4 µM ZR	[125]
Highbush blueberry *Vaccinium corybosum* L.	Bluecrop	Leaf blades	Effect of different CKs (TDZ and ZR) on shoot organogenesis	1 µM TDZ yielded 100% regeneration rate (100%) and a number of shoots (9)	[126]
Highbush blueberry *Vaccinium corybosum* L.	Bluecrop	Shoots	Effect of 0.5 mg/L Z with IBA (0.1, 1, and 5 mg/L) on shoot organogenesis	Maximum shoots (3) were produced from 0.5 mg/L Z (with 0.1 mg/L IBA) treatment	[124]
Highbush blueberry *Vaccinium corybosum* L.	Bluejay-83	Nodal segments	Effect of different CKs (iP and Z) on shoot initiation (%) with either low light or dark condition	93% shoot initiation rate occurred in 4 mg/L Z treatment under dark condition	[127]
Highbush blueberry *Vaccinium corybosum* L.	Burlington	Nodal segments	Effect of different CKs (iP and Z) on shoot initiation (%) with either low light or dark condition	4 mg/L Z resulted in 100% shoot initiation rate under low light condition	[127]
Highbush blueberry *Vaccinium corybosum* L.	Cabot	Nodal segments	Effect of different CKs (iP and Z) on shoot initiation (%) with either low light or dark condition	Highest shoot initiation rate (78%) was obtained from 10 mg/L iP treatment under low light condition	[127]
Highbush blueberry *Vaccinium corybosum* L.	Coville	Nodal segments	Effect of different CKs (iP and Z) on shoot initiation (%) with either low light or dark condition	Low light conditions and 4 mg/L Z resulted in high shoot initiation rate (73%)	[127]
Highbush blueberry *Vaccinium corybosum* L.	Emerald	Nodal segments	Effect of different CKs (TDZ, Z, and ZR) alone or in combination with NAA (2.69 µM) on shoot organogenesis	9.08 µM TDZ improved plant regeneration rate (82%) and shoot number (13)	[125]
Highbush blueberry *Vaccinium corybosum* L.	Evelyn	Nodal segments	Effect of different CKs (iP and Z) on shoot initiation (%) with either low light or dark condition	80% shoot initiation rate was observed in 4 mg/L Z under low light condition	[127]
Highbush blueberry *Vaccinium corybosum* L.	Goldtraube	Shoots	Effect of 0.5 mg/L Z with IBA (0.1, 1, and 5 mg/L) on shoot organogenesis	Maximum shoots (2) were produced in 0.5 mg/L Z (with 1 mg/L IBA) treatment	[124]
Highbush blueberry *Vaccinium corybosum* L.	Herbert	Nodal segments	Effect of different CKs (iP and Z) on shoot initiation (%) with either low light or dark condition	Shoot initiation rate (89%) increase at 4 mg/L Z treatment under low light condition	[127]
Highbush blueberry *Vaccinium corybosum* L.	Jewel	Nodal segments	Effect of different CKs (TDZ, Z, and ZR) alone or in combination with NAA (2.69 µM) on shoot organogenesis	Regeneration rate (89%) and number of shoots (13) were high in treatments consisting of 4.54 µM TDZ with 2.69 µM NAA	[125]
Highbush blueberry *Vaccinium corybosum* L.	Jubilee	Nodal segments	Effect of different CKs (TDZ, Z, and ZR) alone or in combination with NAA (2.69 µM) on shoot organogenesis	4.54 µM TDZ with 2.69 µM NAA treatments yielded high regeneration rate (53.3%) and number of shoots (3)	[125]
Highbush blueberry *Vaccinium corybosum* L.	Northsky	Nodal segments	Effect of different CKs (iP and Z) on shoot initiation (%) with either low light or dark condition	Shoot initiation rate (71%) was highest in 4 mg/L Z treatment under low light condition	[127]
Highbush blueberry *Vaccinium corybosum* L.	O’Neal	Nodal segments	Effect of different CKs (iP and Z) on shoot initiation (%) with either low light or dark condition	88% shoot initiation rate was observed at 15 mg/L iP treatment under dark condition	[127]
Highbush blueberry *Vaccinium corybosum* L.	Pemberton	Nodal segments	Effect of different CKs (iP and Z) on shoot initiation (%) with either low light or dark condition	Highest shoot initiation rate (83%) occurred in 4 mg/L Z treatment under low light condition	[127]
Highbush blueberry *Vaccinium corybosum* L.	Pioneer	Nodal segments	Effect of different CKs (iP and Z) on shoot initiation (%) with either low light or dark condition	4 mg/L Z increased shoot initiation rate (63%) under low light conditions	[127]
Highbush blueberry *Vaccinium corybosum* L.	Washington	Nodal segments	Effect of different CKs (iP and Z) on shoot initiation (%) with either low light or dark condition	Shoot initiation rate (85%) improved in explants treated with 4 mg/L Z under low light condition	[127]
Highbush blueberry *Vaccinium corymbosum* L	Ozarkblue	Leaves	Effect of different CKs on shoot proliferation	Maximum shoots (20) were produced at 20 µM Z treatment	[128]
Highbush blueberry *Vaccinium corymbosum* L.	Duke	Nodal segments	Effect of CKs (TDZ, Z, and iP) on shoot proliferation	2 mg/L Z yielded maximum shoots (4)	[96]
Highbush blueberry *Vaccinium corymbosum* L.	Sunrise	Leaves	Effect of CK conjugates (ZR at 10, 20, and 30 μM) in comparison to Z (10, 20, or 30 μM) and iP (15 μM) on shoot proliferation from leaf section explants	Highest number (21 shoot/explant) of shoot occurred in 20 µM ZR treatment (six times higher than 15 μM iP)	[129]
Highbush blueberry *Vaccinium corymbosum L.*	Bluecrop	Leaves	Effect of CK conjugates, ZR (10, 20, and 30 μM) in comparison to Z (10, 20, or 30 μM) and iP (15 μM) on shoot proliferation from leaf section explants	No shoot regenerated in all treatments	[129]
Highbush blueberry *Vaccinium corymbosum L.*	Duke	Leaves	Effect of CK conjugates, ZR (10, 20, and 30 μM) in comparison to Z (10, 20, or 30 μM) and iP (15 μM) on shoot proliferation from leaf section explants	No shoot regenerated in all treatments	[129]
Highbush blueberry *Vaccinium corymbosum* L.		Double-node segments	Effect of different light treatment with/without 1 mg/L ZR on shoot proliferation	A combination of 100% red LEDs with ZR treatment improved shoot number, number of nodes, leaf number, and leaf area. Application of ZR significantly enhanced shoot proliferation	[130]
Bog bilberry *Vaccinium uliginosum* L.		Young nodal segments	Effect of Z, 2iP, TDZ, IBA, IAA, and GA on shoot multiplication	Maximum shoot number and shoot length were obtained from a combination of Z (2.0 mg/L), 0.1 IBA (mg/L), and 0.2 GA (mg/L)	[131]
Lingonberry *Vaccinium vitis-idaea* L.	Red Pearl	Leaves	Effect of different CKs on shoot proliferation	Maximum shoots (30) produced in 20 µM Z (with 1 µM NAA) treatment	[128]
Lingonberry *Vaccinium vitis-idaea* L. ssp. *minus* (Lodd.) Hult.		Immature leaves	Development and standardization of shoot regeneration protocol in a liquid culture medium using bioreactor systems and solid medium with PGRs (9.1 μM Z and 1.8 μM TDZ)	Shoot multiplication was 2–3 times better in liquid medium than on a semi-solid medium. Z produced vigorous and longer shoots that had more leaves per shoot	[132]
Grape rootstocks *Vitis champini*	Dogridge	Single-node segments	Effect of BA (2–4 mg/L), KIN (2–4 mg/L) individually or in combination with 0.2 mg/L NAA on culture initiation. Shoot proliferation/rooting were established with IBA (2 or 4 mg/L) individually or in combination with activated charcoal (200 mg/L)	BA (2) + NAA (0.2) improved culture initiation (55.98%) and time taken for buds to sprout (6.3 days) in the genotype. 72.35% plantlet survival following IBA and activated charcoal treatment	[133]
Grape *Vitis vinifera* L.	‘Red Globe’	Nodal segment -	Effect of subculturing intervals, BAP (1, 2, and 3 mg/L), Ca concentrations (120.12, 180.18, and 240.24 mg/L), boron (1.08 mg/L), and boric acid (1.08, 2.17, and 3.25 mg/L) at full and ½ MS medium on STN disorder	STN disorder can be best managed in ½ MS media with 1 mg/L BAP, 180.18 mg/L Ca, and 1.08 mg/L boric acid with a 2-week subculture interval	[134]
Grape rootstocks *Vitis vinifera* × *Vitis labrusca*	H-144	Single-node segments	Effect of BA (2–4 mg/1), KIN (2–4 mg/1) individually or in combination with 0.2 mg/1 NAA on culture initiation. Shoot proliferation/rooting were established with IBA (2 or 4 mg/1) individually or in combination with activated charcoal (200 mg/L)	Culture establishment was relatively low (38.31%) with BA (2) + NAA (0.2), and bud sprouting took 14.3 days. Rooting was increased with IBA and charcoal, resulting in 87.75% plantlet survival	[133]

2,4-D = 2,4-dichlorophenoxyacetic acid; 4-CPPU = N-(2-chloro-4-pyridyl)-N-phenylurea; BA = *N*^6^-benzyladenine; BAP = 6-benzylaminopurine; BAR = *N*^6^-benzyladenine riboside; CK = cytokinins; GA = gibberellins; IBA = indole-3-butyric acid; iP = *N*^6^-(2-isopentenyl)adenine; KIN = kinetin; *m*T = *meta*-topolin; *m*TR = *meta*-topolin riboside; Me*m*T = *meta*-methoxy topolin; Me*m*TR = *meta*-methoxy topolin riboside; *m*TTHP = *meta*-topolin tetrahydropyran-2-yl; NAA = naphthalene acetic acid; PGRs = plant growth regulators; STN = shoot-tip necrosis; TDZ = thidiazuron; Z = zeatin; ZR = zeatin ribosides.

**Table 3 biomolecules-10-01222-t003:** Cytokinin applications in mass propagation of major horticultural fruit species through somatic embryogenesis.

Species	Cultivar/Accession	Source of Explants	Factor/s Investigated	Major Outcomes	Reference
Papaya *Carica papaya* L.	Maradol Roja	Immature zygotic embryos (hermaphrodite flowers)	Effect of culture conditions on the germination of somatic embryos: RITA^®^ temporary immersion system. ½ MS + vitamins + BAP + GA_3_	Optimum inoculum density of 200 mg fresh mass of somatic embryos produced 95% germination of somatic embryos	[135]
Sweet orange *Citrus sinensis* (L.) Osbeck	Washington Navel	Pistils from flower buds (20-year-old plants)	Effect of BAP, NAA, and 2,4-D on somatic embryo induction and conversion into plantlets. MS medium + Nitsch and Nitsch vitamins	Induction of embryogenic calli; germination of somatic embryos into plantlets; in vitro grafting	[136]
Sweet orange *Citrus sinensis* Osbeck.)	Tobias	Pistils from flower buds (5-year-old plants)	Effect of BAP, NAA, and 2,4-D on somatic embryo induction and conversion into plantlets. WPM, N6 medium	Induction of embryogenic calli; germination of somatic embryos into plantlets; in vitro grafting	[137]
Litchi *Litchi chinensis* Sonn.	Brewster; Mauritius	Leaves of new flushes of mature (>100-year-old) trees	Induction of embryogenic cultures from leaves of mature trees. B5 Gamborg salts + MS vitamins + 2,4-D + BA, KIN, Z	Induction of embryogenic cultures, maturation of somatic embryos, recovery of plants, and greenhouse acclimatization of plants	[138]
Litchi *Litchi chinensis* Sonn.	Xiafanzhi	Immature zygotic embryos	Isolation of protoplasts from embryogenic suspensions and formation of somatic embryos from protoplasts. MS salts + B5 Gamborg vitamins + KIN + NAA (SE development); KIN + GA₃ (conversion to plantlets)	Development of somatic embryos from protoplast-derived proembryos and their regeneration to plantlets	[139]
Apple *Malus* × *domestica* Borkh.	Golden delicious	Cotyledons of immature zygotic embryos	Effect of PGR combination on somatic embryogenesis. MS + combinations of PGRs: BAP, KIN, NAA, and IBA	Induction of embryogenic masses (18%); conversion of somatic embryos to regenerants (30%)	[83]
Apple *Malus* × *domestica* Borkh.	‘Gloster 69’	Cotyledon-derived cultures of immature zygotic enmbryos	Influence of different PGR combinations on secondary somatic embryogenesis. MS + combinations of PGRs: NAA, IBA, 2,4-D, KIN, and BAP or with TDZ	Optimum SSE (>73%) culture of large size solrtatic embryos or cotyledon-like structures on medium containing a combination of NAA/BAP/KIN or TDZ (10 btM) alone	[140]
Banana *Musa* spp.	Grand Naine (AAA); Tropical (AAAB)	Immature male flowers	Effect of BAP and NAA on the induction of somatic embryos. MS medium + glutamine (0–200 mg/L)	Induction of embryogenic cultures in cell suspension culture; somatic embryo formation and conversion to plantlets	[141]
Banana *Musa* spp.	Chenichampa (AAB)	Immature female flower bud	Effect of 5 different MS medium on the induction of somatic embryos MS 1–5 contained PGRs (IAA, TDZ, NAA, Z, 2iP, BAP), biotin, L- ascorbic acid	MS-3 (0.10 mg/L NAA, 0.3 mg/L Z, 0.14 mgl/ 2iP) supplemented with CH:Gln (100:150 mg/L) increased somatic embryo formation tenfold relative to the control	[142]
Passion fruit *Passiflora edulis* Sims	FB-100-Magaury; FB-200-Yellow master; FB-300	Mature zygotic embryo	Effect of 2,4-D and BA on somatic embryo formation. MS medium + B5 vitamins	Induction of embryogenic calli; germination of somatic and differentiation of embryos; no conversion of embryos into plantlets	[143]
Passion fruit *Passiflora edulis* Sims	FB-300	Mature zygotic embryos	Effect of picloram, BA, and TDZ on somatic embryo formation	Formation of somatic embryos confirmed by histochemical tests	[144]
Avocado *Persea americana* Mill.	Duke 7	Immature zygotic embryos	Influence of semi-permeable cellulose acetate membranes on SE maturation and germination. SE induction: MS medium + picloram; SE germination: MS + BA + GA₃	Increase in SE germination rate: 10% (control) to 40% when SEs were prematured on cellulose acetate membranes	[145]
Avocado *Persea americana* Mill.	Duke 7	Immature zygotic embryos	Effect of BA + GA₃ on the germination of transgenic embryogenic masses. MS + Gamborg macronutrients	Significant improvement in the germination of transgenic somatic embryos on MS medium + BA + GA₃	[146]
Avocado *Persea americana* Mill.	Duke 7; Reed; Haas; A10	Immature zygotic embryos	Developing an efficient plant regeneration system from somatic embryos. SE induction: Gamborg’s B5 major salts + MS minor salts + picloram; SE maintenance: Gamborg’s B5 major salts + MS minor salts + MS vitamins + 2,4-D; SE germination: MS salts + BA + GA₃	Total plant regeneration (58.3%, including 43.3% bipolar regeneration) and 36.7% plant recovery rate	[147]
Avocado *Persea americana* Mill.	Duke 7	Immature zygotic embryos	Establish a successful plant recovery system from transgenic somatic embryos. SE induction: MS medium + picloram; plant recovery: MS + TDZ + BA; micrografting: MS + BA	Micrografts on MS + BA allowed a 60–80% success rate	[148]
Pomegranate *Punica granatum* L.	NS	Immature zygotic embryos and cotyledons (unripen fruits)	Somatic embryo formation and plant regeneration. MS + 2,4-D + BAP + KIN	Proliferation of embryogenic cell clusters, embryo maturation, and the production of young seedlings	[149]
Guava *Psidium guajava* L.	Allahabad Safeda; Lalit; Sardar (L-49); Shweta	Immature zygotic embryos	Effect of BAP and NAA on the germination of somatic embryos induced on 2,4-D; multiplication of plantlets on BAP	High genetic fidelity of regenerated plantlets – 99% exhibited monomorphic bands using RAPD, ISSR, and SSR markers	[82]
Guava *Psidium guajava* L.	Carabao	Nucellar tissue of immature fruits	Effect of 2,4-D, BAP, and KIN on multiplication of germinated somatic embryos	Regenerated plantlets survived *ex vitro* conditions but at low survival frequency	[150]
Grape *Vitis vinifera* L.	Thompson Seedless	Immature stamens	Long-term culture system for maintenance and transformation MS + 2,4-D + BAP; KIN	Somatic embryos could be propagated over 3 years and were amenable to *Agrobacterium*-mediated transformation	[151]
Grape *Vitis vinifera* L.	Chardonnay	Nodal explants of in vitro plantlets	Somatic embryo production from nodal explants. ½ MS + 2,4-D + BAP	Nodal explants produced SEs but intermodal segments, petioles, and leaf segments were not amenable to somatic embryogenesis under similar conditions	[79]
Grape *Vitis vinifera* L.	Chardonnay, Müller Thurgau, Grignolino and Brachetto a grappolo lungo	Immature whole flowers, anthers, ovaries	Initiating somatic embryogenesis in grapevine from immature whole flower explants. Nitsch and Nitsch salts + MS vitamins + 2,4-D + BA	No morphological differences between embryogenic cultures from anthers, ovaries, and whole flowers	[152]
Grape *Vitis vinifera* L.	Mencía	Immature stamens (anther+ filament) and ovaries from adult grown plants	Embryogenic competence of ovaries and stamens, and effect of 2,4-D; TDZ on embryogenic response Nitsch and Nitsch salts + MS vitamins + 2,4-D + TDZ	High SE germination (87%) and plant conversion rate (88%)	[153]
Grape *Vitis vinifera* L.	Chardonnay	Immature anthers, ovaries and whole flowers	Develop a system for embryogenic culture induction, maintenance, and transformation. Embryogenic culture: MS + picloram + 2,4-D + BA Embryo germination: MS + KIN + NOA	Establishment of a picloram-induced system for embryogenic tissue initiation, maintenance, proliferation, regeneration, and *Agrobacterium*-based transformation	[154]

2,4-D = 2,4-dichlorophenoxyacetic acid; 2iP = 2-isopentenyladenine; BA = *N^6^*-benzyladenine; BAP = benzylamino purine; CH = casein hydrochloride; GA₃ = gibberellic acid; Gln = glutamine; KIN = kinetin; IAA = indole-3-acetic acid; IBA = indole-3-butyric acid; MS = Murashige and Skoog medium; NAA = naphthalene acetic acid; NOA = 2-naphthoxyacetic acid; PGRs = plant growth regulators; SE = somatic embryo; TDZ = thidiazuron; WPM = woody plant medium; Z = Zeatin. NS = not specified.

**Table 4 biomolecules-10-01222-t004:** Fruit attribute responses following preharvest cytokinin application in horticultural fruit crops.

Species	Cultivar	Factor(s) Investigated	Major Outcome(s)	Reference
Hardy kiwifruit *Actinidia arguta* (Sieb. et Zucc.) Planch. ex Miq.	Mitsuko	Effect of application time (0, 1, 10, and 25 DAPF) on the fruit size using 5 mg/L CPPU	Highest fruit size (16 vs. 7 g in control) occurred in CPPU treatment applied at 10 DAPF	[214]
Hardy kiwifruit *Actinidia arguta* (Sieb. et Zucc.) Planch. ex Miq.	Mitsuko	Effect of CPPU (0, 1, 5, and 10 mg/L) on the fruit size and quality	Fruit growth was markedly enhanced in CPPU (1–5 mg/L) treatment	[214]
Kiwifruit *Actinidia chinensis* var. *deliciosa*	Hayward	Effects of CPPU on fresh fruit quality under standard and high carbohydrate supply	CPPU decreased fruit dry matter due to increased fruit growth caused by elevated glucose and fructose levels during the early stage of fruit growth grown under high carbohydrate supply	[56]
Kiwifruit *Actinidia deliciosa* (A. Chev.) C.F. Liang and A.R. Ferguson	Hayward	Effect of CPPU (0, 10, and 20 mg/L) sprayed at two different application times (14 and 35 DAFB) on productivity, fruit quality, and storage life	CPPU increased fruit size and yield per vine at all concentrations and application times tested, without affecting fruit seed number and fruit drop. Late treatment (35 DAFB) increased fruit size and yield by promoting water accumulation, with negative effects on the qualitative characteristics	[215]
Kiwifruit *Actinidia deliciosa* (A. Chev.) C.F. Liang and A.R. Ferguson	Hayward with cv. Matua as pollinizer (6:l)	Effect of CPPU (0 and 20 mg/L) on productivity	About 25% higher yield per vine in CPPU treatment than the control	[216]
Kiwifruit *Actinidia deliciosa* (A. Chev.) C.F. Liang and A.R. Ferguson var. *deliciosa*	Hayward	Effect of interaction between date of anthesis (early and late flowering) and CPPU (0 and 15 µl/l) on cell number and size as well as final weight of the fruits	CPPU-treated fruit from two bloom dates had higher cell numbers in the outer pericarp at harvest. However, the cell size in the inner pericarp of early and late untreated fruits was higher than that of CPPU-treated fruit. The early flowers had larger commercial fruit size (153 g) than those from later flowers (126 g)	[217]
Kiwifruit *Actinidia deliciosa* (A. Chev.) C.F. Liang and A.R. Ferguson *var*. deliciosa	Hayward	Effect of the interaction between date of anthesis (early or late) and CPPU (15 µl/l) on cell number and size as well as the final weight of the fruit	CPPU in the early flowers had a much larger commercial fruit size (153 g) than fruit from later flowers (126 g). CPPU-treated fruits from the two bloom dates had higher cell number in the outer pericarp, while the cell size in the inner pericarp of early and late untreated fruits was higher than that in CPPU-treated fruit	[217]
Kiwifruit *Actinidia deliciosa* (A. Chev.) C.F. Liang and A.R. Ferguson	Hayward	Comparison of fruit weight at harvest and number of viable seeds in pollinated and CPPU-treated (40 mg/L) parthenocarpic fruit	Parthenocarpic fruit were heavier than pollinated fruit. Majority of the aborted seeds were found in parthenocarpic fruit	[218]
Kiwifruit *Actinidia deliciosa* (A. Chev.) C.F. Liang and A.R. Ferguson	Hayward	Effect of CPPU, 2,4-D, and GA_3_ alone or in combination on diameter, length, and firmness, as well as fresh and dry weight of fruits	CPPU + GA_3_ + 2,4-D and CPPU + 2,4-D treatments recorded increase in fruit length and diameter. CPPU + GA_3_ + 2,4-D treatment increased the number of larger fruits	[219]
Pineapple *Ananas comosus* (L.) Merr.	N-67-10	Effect of BA (0, 10, 25, and 50 mg/L) and CPPU (0, 1.25, 2.5, and 5 mg/L) on the growth of stem cutting	Production of about 21 buds per stem after treatment with 5 mg/L CPPU solution for more than 3 h. More than 18 (89%) of the buds eventually sprouted	[220]
Watermelon *Citrullus lanatus* Thunb.	Reina de corazones	Effect of CPPU (0, 50, 100, 150, and 200 mg/L) and 2,4-D (0, 4, 6, 8, and 12 mg/L) on productivity	Production and number of fruit obtained with CPPU treatments were similar to those that were bee pollinated. Maximum production was reached at 100–200 mg/L	[221]
Persimmon fruit *Diospyros kaki* Thunb.	Triumph	Effect of Superlon (mixture of GA_4+7_ and BA; 0 and 40 µg/mL) applied once a month (from 40 DAFS) for three consecutive months on growth responses	Treated fruits at either 40 or 100 DAFS showed inhibition of fruit growth by 3% or 6%, respectively	[222]
Japanese persimmon *Diospyros kuki* Thunb.	Matsumoto-Wase-Fuyu	Effect of CPPU (0, 5, and 10 mg/L) applied 11 DAFB on the number, size, and yield of marketable fruits	Number and yield of marketable fruits from 5 mg/L CPPU treatment was slightly higher than the control. CPPU (10 mg/L) had a significantly bigger size than the control, while the number of seeds was similar between CPPU treatments and control	[223]
Strawberry *Fragaria × ananassa* Duch.	Akihime	Effect of CPPU (0, 5, 10, and 15 mg/L) preharvest treatment on fruit quality	High doses of CPPU (10 and 15 mg/L) increased fresh weight	[55]
Apple *Malus domestica* Borkh.	Fuji	Effect of CPPU (10 mg/L), CPPU (10 mg/L) + GA_4+7_ (100 mg/L) on fruit weight, length, and width at 4 and 30 DAFB	Highest fresh weight, length, diameter, and L/D ratio was observed with the use of CPPU (10 mg/L) applied at 4 DAFB	[211]
Apple *Malus domestica* Borkh.	Fuji	Effect of different concentrations of CPPU (1, 5, and 10 mg/L) on fruit weight, length, and width at 10, 20, and 30 DAFB	Highest fresh weight, length, and diameter was observed with the use of 5 mg/L CPPU applied at 4 DAFB	[211]
Apple *Malus domestica* Borkh.	Oregon Spur Delicious/MM.111	Effect of CPPU (0, 6.25, 12.5, 25, and 50 mg/L) and GA on fruit quality (L/D ratio and flesh firmness and weight) at FB and 14 DAFB	CPPU (12.5–25 mg/L) increased the fruit L/D ratio. Flesh firmness increased linearly with an increase in concentration. CPPU applied at FB or 14 DAFB did not affect fruit weight	[224]
Apple *Malus domestica* Borkh.	Delicious/M.26	Effect of CPPU (0, 5, 10, 15, and 20 mg/L) on seed number and fruit quality (L/D ratio, flesh firmness and weight)	L/D ratio increased in all treatments compared to the control. Flesh firmness at harvest linearly increased with increasing concentrations. CPPU treatment had no significant stimulatory effect on fruit weight and seed number	[224]
Apple *Malus domestica* Borkh.	Empire/MM.106	Effect of CPPU (0, 10, 20, and 40 mg/L) on fruit quality (L/D ratio, diameter, flesh firmness, and weight)	CPPU treatment increased fruit weight linearly but did not affect fruit set, L/D, and firmness	[224]
Apple *Malus domestica* Borkh.	Early McIntosh/M.7	Effect of CPPU (50 and 100 mg/L) and NAA (20 mg/L) on the number of fruits and percentage of drop	CPPU (100 mg/L) reduced the fruit drop by 47% when compared to the control at 6 DAT. However, this effect diminished after 14 days.	[224]
Apple *Malus domestica* Borkh.	Hi-Early Delicious	Effect of promalin and CPPU on the number of blossom and fruit quality (circumference and flesh firmness)	CPPU treatment increased the red surface (%) of fruit. Treatments had no significant effect on return bloom. Fewer culls due to poor color in the CPPU-treated fruits	[224]
Apple *Malus domestica* Borkh.	Empire	Effect of BA (0, 50, 100, and 150 mg/L) applied 13 DAFB on crop load and shoot growth	Fruit set and fruit count per tree were reduced by higher BA concentrations. Yield per tree was unaffected by BA. Mean fruit weight was increased substantially by BA	[225]
Apple *Malus domestica* Borkh.	Empire	Effect of BA (0, 125, and 250 mg/L), TDZ (0, 62, and 125 mg/L) and ethephon (0, 125, and 250 mg/L) applied 22 DAFB on crop load and shoot growth	Fruit set and fruit count per tree decreased by a higher concentration of each chemical. Yield per tree decreased and mean fruit weight increased with BA and TDZ treatments	[225]
Apples *Malus domestica* Borkh.	McIntosh	Effect of TDZ (0, 10, or 50 mg/L) on fruit set, fruit quality (weight, L/D ratio, and flesh firmness) and return bloom at FB and 22 DAFB	TDZ significantly reduced fruit set and seed count. Although crop load was reduced with application at petal fall, the highest reduction occurred with TDZ at 22 DAFB. TDZ (especially 50 mg/L) reduced return bloom at 22 DAFB more than at FB. Fruit weight and L/D ratio were increased by the FB spray. 22 DAFB reduced the red surface (%) while fruit from either FB or 22 DAFB were irregularly shaped. Flesh firmness increased with 50 mg/L at 22 DAFB	[226]
Apples *Malus domestica* Borkh.	Empire	Effect of TDZ (0, 1, 5, and 15 mg/L) on fruit set, fruit quality, and return bloom at FB and 18 DAFB. In addition, the number of fruits with bitter pit was noted	Although TDZ increased fruit weight when applied at FB or 18 DAFB, the later application was less effective. TDZ increased firmness when applied at FB but decreased firmness at 18 DAFB. SSC increased only when TDZ was applied at 18 DAFB. The fruit L/D ratio increased at both times of application, but the highest increase occurred with the FB treatment. No treatment influenced the red surface (%) or bitter pit and return bloom	[226]
Apples *Malus domestica* Borkh.	Double Red Delicious	Effect of CPPU, TDZ (0, 5, and 10 mg/L) and promalin (25 mg/L) on fruit set, quality, and return bloom	All the treatments (CPPU > TDZ > promalin) increased the fruit L/D ratio. TDZ and CPPU increased flesh firmness. Relative to the control, no treatment improved the fruit weight and seed count. TDZ and CPPU (10 mg/L) reduced return bloom	[226]
Apples *Malus domestica* Borkh.	Marshall McIntosh	Effect of timing and CPPU (0 and 8 mg/L) application on fruit set, fruit characteristics, and return bloom	CPPU caused fruit thinning at 5 and 10 mm of fruit size. CPPU increased the return bloom on trees that it thinned the previous year, but slightly decreased bloom when applied for fruit size ranging from 16 to 22 mm. CPPU significantly increased fruit size, but the maximum response was noticed when applied between petal fall and 10 mm	[227]
Apples *Malus domestica* Borkh.	McIntosh	Effect of CPPU (0, 1, 2, 4, and 8 mg/L) applied at 5–6 mm on fruit set and return bloom	Increasing concentration of CPPU increased the fruit weight, fruit flesh firmness, and the percentage of asymmetrical fruit, while a linear reduction in fruit set and seed number was observed. No treatment improved the return bloom and red color of the fruit	[227]
Apples *Malus domestica* Borkh.	McIntosh	Effect of the time of application (petal fall or 5–6 mm) and concentrations (0, 2, 4, or 6 mg/L) of CPPU on fruit set and return bloom	CPPU caused a linear increase in fruit weight. Increasing the concentration of CPPU reduced the fruit set, seed number, and a delay in ripening	[227]
Apples *Malus domestica* Borkh.	Starkrimson Delicious	Effect of BA (0 and 75 mg/L), NAA (0 and 6 mg/L), and carbaryl (0 and 600 mg/L) used alone or in combination on fruit set and return bloom	All compounds reduced the number of seeds when expressed on a per limb cross-sectional area basis. BA and carbaryl enhanced return bloom. BA and carbaryl resulted in heavier fruit than in the control	[228]
Apples *Malus domestica* Borkh.	Redspur Delicious	Effect of BA (0 and 75 mg/L), NAA (0 and 6 mg/L), and carbaryl (0 and 600 mg/L) used alone or in combination on fruit set and return bloom	All compounds reduced seed counts, but only carbaryl and BA increased return bloom. BA and carbaryl resulted in heavier fruit than in the control	[228]
Apples *Malus domestica* Borkh.	Pioneer McIntosh	Effect of BA (0 and 150 mg/L) and ABA (0, 50, 150, 300 and 1000 mg/L) on fruit set and fruit quality	BA increased the return bloom and mean fruit weight. Fruit weight from trees treated with both ABA and BA was similar in weight to those sprayed with BA alone	[209]
Apples *Malus domestica* Borkh.	Marshall McIntosh	Effect of BA (0 and 150 mg/L) and ABA (0, 50, 150, 300, and 1000 mg/L) on fruit set	Application of ABA and BA resulted in higher thinning and fruit weight. BA increased return bloom compared with the control. BA also increased the fruit flesh firmness	[209]
Apples *Malus domestica* Borkh.	Autumn Rose Fuji	Effect of BA (0 and 150 mg/L) and ABA (0, 50, 150, 300, and 1000 mg/L) on fruit set	BA and ABA reduced final fruit set, and the thinning interaction between the two was significant. BA increased the fruit L/D ratio	[209]
Apples *Malus domestica* Borkh.	Morespur McIntosh	Effect of BA (0, 50, and 100 mg/L) on fruit thinning, seed number of abscising and persisting fruit	BA significantly increased the fruit weight but not the number of seeds and fruits	[229]
Apples *Malus domestica* Borkh.	Anna	Effect of CPPU, Inca, Calbor, humic acid, and their combination on fruit set, yield, storage quality, and reduced number of fruit decay at harvest and preharvest	CPPU individually or in combination with Inca, Calbor, or humic acid increased fruit set, branch number, fruit drop, fruit number, and yields	[230]
Apples *Malus domestica* Borkh.	Fuji	Effect of BA (300 mg/L) on flower bud formation	BA increased the flowering rate and shoot proportion in the spur clusters	[58]
Mango *Mangifera indica* L.	Kesha	Effect of CPPU (10 and 20 mg/L) and NAA (40 mg/L) on the yield of mango following applications at mustard, pea, and marble stages	CPPU (10 mg/L applied at mustard and pea stages) treatment had the maximum number of fruit per panicle at pea stage (14.54), marble stage (4.39), and fruit harvested per panicle (1.52) as well as the yield of fruit (107 kg/tree and 10.7 tonnes/ha). In addition, 20 mg/L CPPU (applied at marble + pea stages) increased the fruit length (10.56 cm), diameter (6.43 cm), and average fruit weight (328.73 g)	[231]
Mango *Mangifera indica* L.	Alphonso	Effects of different PGRs (e.g., NAA-20 mg/L, CPPU-10 mg/L, paclobutrazol-10 and 25 mg/L) on fruit yield	CPPU significantly increased the number of fruit/tree and yield when compared to the control. CPPU increased (143.5%) the number of fruits retained at harvest	[232]
Sweet cherry fruit *Prunus avium* L.	Bing	Effect of BA, CPPU, *m*T, TDZ, GA (GA_1_, GA_3_, GA_4+7_) applied to fruit pedicels at 9 or 30 DAFB on fruit quality	CKs applied 30 DAFB improved fruit weight significantly (about 15% increase) with CPPU and *m*T at 100 mg/L being the most effective treatments	[204]
Sweet cherry fruit *Prunus avium* L.	Bing	Effect of five PGRs (CPPU, GA_1_, GA_3_, and GA_4+7_, fluridone) applied alone or combination during pit hardening on fruit weight and firmness	CPPU did not increase fruit weight or pit weight but reduced seed growth and induced about 85% aborted seeds	[233]
Pear *Pyrus communis* L.	Akca	Effect of BA (0, 50, 100, and 150 mg/L) and GA_4+7_ (0, 12.5, 25, and 50 mg/L) on fruit size and quality (fruit weight, size, and fruit color)	BA (100 mg/L) significantly improved fruit size with no negative effect on the yield and fruit shape. The heaviest and longest fruits were obtained from 25 and 50 mg/L BA + GA_4+7_	[205]
Pear *Pyrus communis* L.	Spadona	Effect of BA on fruit size, thinning, and yield	BA increased the fruit size and yield without any influence on fruit shape and seed number. Slight thinning was also observed	[206]
Pear *Pyrus communis* L.	Coscia	Effect of BA on fruit size, thinning, and yield	BA increased the fruit size and yield without any influence on fruit shape and seed number. A heavy thinning effect was observed	[206]
Pear *Pyrus communis* L.	Spadona	Effect of CPPU (0, 10, and 20 mg/L) on fruit size, quality, and shape in relation to timing (7, 14, 21, and 28 DAFB) during 1997 field trials	CPPU (20 mg/L) significantly increased fruit size when applied at 7, 14, and 21 DAFB. At harvest, the total increase in the yield of large fruits was 100% and 170% for CPPU 10 and 20 mg/L, respectively. CPPU-treated fruits had a bigger diameter	[234]
Pear *Pyrus communis* L.	Spadona	Effect of CPPU (0, 2.5, 5, 10, and 20 mg/L) on fruit size, quality, and shape after 14 DAFB application field trials in 1999	CPPU (< 10 mg/L) did not affect fruit size while 20 mg/L significantly increased the yield of large fruits	[234]
Pear *Pyrus communis* L.	Costia	Effect of CPPU (0 and 20 mg/L) on fruit size, quality, and shape in relation to timing (7, 14, and 21 DAFB) during 1997 field trials	CPPU treatments significantly increased fruit size when compared to the control. At harvest, the total increase in the yield of large fruits in CPPU treatment was 150–200% (13: 16 vs. 5 tonnes/ha in control)	[234]
Pear *Pyrus communis* L.	Bartlett	Effect of BA (0 and 150 mg/L) and ABA (0 and 250 mg/L) on fruit set and quality	Even though BA reduced fruit set numerically by at least 35%, the reduction was not significant. BA had no influence on fruit quality parameters at harvest except for slightly increasing the extent of fruit russeting	[235]
Blackberry *Rubus* spp L.	Chester Thornless	Effect of PGRs (for e.g., BA, GA at 0 and 100 mg/L) on primocane growth and architecture	Although BA alone generally had no positive effects, BA + GA increased branch production and elongation as well as the dry weights of some component tissues	[236]
Rabbiteye blueberries *Vaccinium ashei* Reade	Tifblue	Effect of CPPU (0 and 10 mg/L) alone or with GA_3_ applied at different time intervals (7 and 17 DAFL) on the fruit set berry size and mean ripe berry weight in the 1999 growing season	Number of berry (1680 vs. 435 in control) was highest in CPPU (17 DAFL) with GA_3_ treatment. Mean ripe berry weight was higher in CPPU (1.34 g) without GA_3_ treatment compared to control (1.18 g)	[237]
Rabbiteye blueberries *Vaccinium ashei* Reade	Tifblue	Effect of CPPU (0 and 10 mg/L) alone or with GA_3_ applied at different time intervals (7 and 17 DAFL) on the fruit set, berry size, and yield in the 2000 growing season	Number of berry (1680 vs. 435 in control) was highest in CPPU (17 DAFL) with GA_3_ treatment. Fruit set (98%) was highest in CPPU with GA_3_ applied 17 DAFL. Mean ripe berry weight was higher in CPPU (1.33 g) without GA_3_ treatment compared to control (0.99 g). CPPU (17 DAFL) without GA_3_ treatment had the highest yield (8.3 kg/bush)	[237]
Rabbiteye blueberries *Vaccinium ashei* Reade	Brightwell	Effect of CPPU (0 and 15 mg/L) applied at 7, 14, 21, and 28 DAFL on fruit set (%) and mean berry weight under greenhouse conditions	CPPU applied at 7 DAFL had about 50% more fruit set and significantly higher weight (1.6 vs. 1.3 g) than the control	[238]
Rabbiteye blueberries *Vaccinium ashei* Reade	Climax	Effect of CPPU (0 and 15 mg/L) applied at 7, 14, 21, and 28 DAFL on fruit set (%) and average berry weight under greenhouse conditions	CPPU applied at 28 DAFL had about 50% more fruit set and significantly higher weight (1.6 vs. 1.5 g) than the control	[238]
Rabbiteye blueberries *Vaccinium ashei* Reade	Tifblue	Effect of CPPU (0 and 15 mg/L) applied at 0, 7, 14, 21, and 28 DAFL on fruit set (%) and mean berry weight under greenhouse conditions	CPPU applied at 14 DAFL had more than 50% fruit set compared to the control. Highest fruit weight was observed in CPPU applied 21 DAFL (1.5 vs. 1.4 g in control)	[238]
Rabbiteye blueberries *Vaccinium ashei* Reade	Brightwell	Effect of CPPU (0 and 15 mg/L) applied at 0, 7, 14, 21, and 28 DAFL on fruit set (%), mean berry weight, and harvest yield under field conditions in Alapaha, GA	Highest fruit set (45%) occurred in CPPU applied 14 DAFL while only 29% set in control. CPPU (28 DAFL) treatment had the highest fruit weight (1.7 vs. 1.4 g in control). CPPU (14 DAFL) treatment had the highest harvest yield (1870 vs. 1545 kg/ha in control)	[238]
Rabbiteye blueberries *Vaccinium ashei* Reade	Climax	Effect of CPPU on (0 and 15 mg/L) applied at 0, 7, 14, 21, and 28 DAFL on fruit set (%), mean berry weight, and harvest yield under field conditions in Alapaha, GA	Highest fruit set (71%) and weight (1.4 vs. 1.3 g) occurred in CPPU applied 21 DAFL while only 28% set in control. CPPU (21 DAFL) treatment had the highest harvest yield (1301 vs. 732 kg/ha in control)	[238]
Rabbiteye blueberries *Vaccinium ashei* Reade	Tifblue	Effect of CPPU on (0 and 15 mg/L) applied at 0, 7, 14, 21, and 28 DAFL on fruit set (%) and average berry weight under field conditions in Griffin, GA	Highest fruit set (75%) and weight (1.6 vs. 1.4 g) occurred in CPPU applied 7 DAFL while only 50% set in control treatment	[238]
Southern highbush blueberries *Vaccinium corymbosum* hybrids L.	Magnolia	Effect of CPPU (0, 5, and 10 mg/L) applied at (10, 14, and 20 DAFL) and number of applications (1 or 2) on fruit set (%) and mean individual berry weight under greenhouse conditions	Highest fruit set (74%) was observed in 10 mg/L CPPU applied once at 14 DAFL. The biggest fruit (2 g) was also observed with the same treatment	[239]
Southern highbush blueberries *Vaccinium corymbosum* hybrids L.	Reveille	Effect of CPPU (0, 5, and 10 mg/L) applied at (10, 14, and 20 DAFL) and number of applications (1 or 2) on fruit set (%) and mean individual berry weight under greenhouse conditions	Highest fruit set (96%) was observed in 5 mg/L CPPU applied once at 14 DAFL. The biggest fruit (2 g) was also observed with the same treatment	[239]
Southern highbush blueberries *Vaccinium corymbosum* hybrids L.	Star	Effect of CPPU (0, 5, and 10 mg/L) applied at DAFL on fruit set (%) and mean individual berry weight under greenhouse conditions	Highest fruit set (88%) was observed in 10 mg/L CPPU applied 7 DAFL. The biggest fruit (1.7 g) was obtained with 5 mg/L CPPU applied at 14 DAFL	[239]
Southern highbush blueberries *Vaccinium corymbosum* hybrids L.	Legacy	Effect of CPPU (0, 5, and 10 mg/L) applied at DAFL on fruit set (%) and mean individual berry weight under greenhouse conditions	Highest fruit set (81%) was observed in 10 mg/L CPPU applied once at 14 DAFL. The biggest fruit (2.2 g) was obtained with 5 mg/L CPPU applied at 14 DAFL	[239]
Southern highbush blueberries *Vaccinium corymbosum* hybrids L.	Palmetto	Effect of CPPU (0, 5, and 10 mg/L) applied at DAFL on fruit set (%) and mean individual berry weight under greenhouse conditions	All the concentrations of CPPU reduced the fruit set (from 64% in control to 23% in 10 mg/L applied at 14 DAFL). The biggest fruit (1.6 g) was obtained with 5 mg/L CPPU applied at 14 DAFL	[239]
Southern highbush blueberries *Vaccinium corymbosum* hybrids L.	Georgiagem	Effect of CPPU (0 and 10 mg/L) applied at 10–14 DAFL on fruit set (%), berry weight, and ripe fruit (%) in Georgia during 2001	CPPU treatment had higher fruit set (50%) than the control (38%). Both CPPU-treated and control fruits had similar sizes. Upon harvest, CPPU (27%) delayed fruit ripening when compared to the control (35%)	[239]
Southern highbush blueberries *Vaccinium corymbosum* hybrids L.	Palmetto	Effect of CPPU (0 and 10 mg/L) applied at 10–14 DAFL on fruit set (%), berry weight, and ripe fruit (%) in Georgia during 2001	CPPU treatment had higher fruit set (71%) than the control (49%). Both CPPU-treated and control fruits had similar sizes. At harvest, CPPU (53%) delayed fruit ripening when compared to the control (72%)	[239]
Southern highbush blueberries *Vaccinium corymbosum* hybrids L.	Reveille	Effect of CPPU (0 and 10 mg/L) applied at 10–14 DAFL on fruit set (%), berry weight, and ripe fruit (%) in Georgia during 2001	Fruit set was similar for both CPPU and control treatments. CPPU increased berry size by 20% compared with control berries, while fruit ripening was delayed	[239]
Southern highbush blueberries *Vaccinium corymbosum* hybrids L.	Bladen	Effect of CPPU (0 and 10 mg/L) applied at 10–14 DAFL on fruit set (%), berry weight, and ripe fruit (%) in Georgia during 2001	CPPU application reduced fruit set (35%) when compared to the control (50%). CPPU increased berry size by 20% compared with control berries, while fruit ripening was delayed	[239]
Southern highbush blueberries *Vaccinium corymbosum* hybrids L.	Millennia	Effect of CPPU (0, 10, and 15 mg/L) applied 10 DAFB with or without surfactant on fruit set (%) and mean individual berry weight under field conditions in Ware County, GA, during 2005	Highest fruit set (38%) was observed with 15 mg/L CPPU applied without surfactant. CPPU (10 mg/L) treatment with surfactant had the biggest fruit size (2.2 g)	[239]
Southern highbush blueberries *Vaccinium corymbosum* hybrids L.	O’Neal	Effect of CPPU (0, 10, and 15 mg/L) applied 13 DAFB with or without surfactant on fruit set (%) and mean individual berry weight under field conditions in Ware County, GA, during 2005	Highest fruit set (71%) was observed with 10 mg/L CPPU applied without surfactant. CPPU (10 mg/L) treatment with surfactant had the biggest fruit size (1.8 g)	[239]
Southern highbush blueberries *Vaccinium corymbosum* hybrids L.	Bluecrisp	Effect of CPPU (0, 10, and 15 mg/L) applied 13 DAFB with or without surfactant on fruit set (%) and mean individual berry weight under field conditions in Ware County, GA, during 2005	Highest fruit set (51%) was observed with 10 mg/L CPPU applied without surfactant. CPPU (15 mg/L) treatment with surfactant had the biggest fruit size (1.9 g)	[239]
Southern highbush blueberries *Vaccinium corymbosum* hybrids L.	Santa Fe	Effect of CPPU (0, 5, and 10 mg/L) applied at 10, 14, or 20 DAFB on yield per plant and fruit sizes under field conditions in Florida	Treatments had no stimulatory effect on the yield relative to the control. CPPU (10 mg/L) application at 20 DAFB produced the biggest fruit size (1.5 g)	[239]
Southern highbush blueberries *Vaccinium corymbosum* hybrids L.	Star	Effect of CPPU (0, 5, and 10 mg/L) applied at 10, 14, or 20 DAFB on yield per plant and fruit sizes under field conditions in Florida	Highest yield was produced during 14 DAFB application of 5 mg/L CPPU (5259 g/plant to 3656 g/plant in control). CPPU (10 mg/L) application at 20 DAFB produced the biggest fruit size (1.6 g)	[239]
Southern highbush blueberries *Vaccinium corymbosum* hybrids L.	Sharpblue	Effect of CPPU (0, 5, and 10 mg/L) applied at 10, 14, or 20 DAFB on yield per plant and fruit sizes under field conditions in Florida	Yield was slightly lower in all CPPU (especially in 10 mg/L applied 20 DAFB) treatments when compared to the control. CPPU (10 mg/L) application at 20 DAFB produced the biggest fruit size (1.6 g)	[239]
Southern highbush blueberries *Vaccinium corymbosum* hybrids L.	Millennia	Effect of CPPU (0, 5, and 10 mg/L) applied at different time intervals (7 and 20 DAFB) on mean berry fresh weight and cumulative yield from harvest period in 2002	CPPU (10 mg/L) application at 20 DAFB produced the biggest fruit size (1.6 g). CPPU treatment did not improve the yield when compared to the control	[239]
Southern highbush blueberries *Vaccinium corymbosum* hybrid*s* L.	Star	Effect of CPPU (0, 5, and 10 mg/L) applied at different time intervals (7 or 14 DAFB) on mean berry fresh weight and cumulative yield from harvest period in 2002	CPPU (5 mg/L) application at 7 DAFB followed by another 5 mg/L at 14 DAFB produced the biggest fruit size (1.5 g). CPPU did not improve the yield when compared to the control	[239]
Table grapes *Vitis labrusca*	Lakemont (seedless)	Effect of different concentrations (0, 5, 10, or 15 mg/L) of CPPU on fruit development (berry size and cluster parameters) at 14, 28, 42, and 56 DAT	CPPU (10 and 15 mg/L) significantly increased berry size at all measurement dates. At maturity, CPPU (10 or 15 mg/L) significantly increased berry and cluster mass, but the number of berries/clusters and compactness were not different from the control	[240]
Table grapes *Vitis labrusca* L.	Himrod	Effect of CPPU (0, 5, 10, and 15 mg/L) on fruit development (berry size and cluster parameters)	Berry size and firmness significantly increased with increasing concentrations of CPPU. Brix was lower (10–4%) at all CPPU concentrations while the compactness rating and mass (64–76%) of the clusters were significantly higher in CPPU than in the control	[240]
Table grapes *Vitis labrusca* L.	Himrod	Effect of CPPU (0, 5, and 10 mg/L), berry diameter (4, 5, 7, and 9 mm) and time (35, 49 63, and 77 DAFB) of application on fruit development (berry size and cluster parameters)	All concentrations of CPPU applied to the different diameters increased berry size. At maturity, CPPU increased berry mass over control by 30%, 27%, and 23% at 4, 5, and 7 mm diameter, respectively. A similar stimulatory effect was observed for cluster mass, while there was no significant positive effect on cluster compactness	[240]
Table grapes *Vitis labrusca* L.	Venessa (seedless)	Effect of CPPU (0, 5, 10, and 15 mg/L) on fruit development (berry size and cluster parameters) at 14, 28, 42, and 56 DAT	CPPU (10 and 15 mg/L) significantly increased berry size at all measurement dates. At maturity, all concentrations of CPPU significantly increased berry mass, number of berries/clusters, as well as cluster weight and compactness	[240]
Table grapes *Vitis labrusca* L.	Concord (seeded)	Effect of CPPU (0, 5, 10, and 15 mg/L) on fruit development (berry size and cluster parameters) at 14, 28, 42, and 56 DAT	CPPU (10 and 15 mg/L) significantly increased berry size at all measurement dates. At maturity, CPPU treatment had no positive effect on the berry mass, number of berries/clusters as well as cluster weight and compactness	[240]
Table grapes *Vitis labrusca* L.	Niagara (seeded)	Effect of CPPU (0, 5, 10, and 15 mg/L) on fruit development (berry size and cluster parameters) at 14, 28, 42, and 56 DAT	CPPU (10 and 15 mg/L) significantly increased berry size at 14 DAT. At maturity, CPPU treatment had no positive effect on the berry mass, number of berries/clusters as well as cluster weight and compactness	[240]
Grape vine *Vitis vinifera* L.	Thompson seedless	Effect of preharvest sprays of CPPU (0 and 5 mg/L), putrescine, GA, salicylic acid, ethephon, ascorbic acid, and calcium chloride on the yield and berry quality at harvest	CPPU and GA_3_ had significantly higher yield and improved firmness compared with the control. CPPU significantly increased cluster weight, berry weight, and juice content as well as sizes of berries (length and width) when compared to the control	[241]
Grapes *Vitis vinifera* L.	Flame Seedless	Effect of CPPU (0, 1, 2, and 3 mg/L) on berry size	CPPU (2 mg/L) significantly increased berry diameter when compared to the control	[242]
Grapes *Vitis vinifera* L.	Redglobe	Effect of CPPU (0, 3, 4, and 5 mg/L) on berry size	CPPU (5 mg/L) increased berry diameter	[242]
Grapes *Vitis vinifera* L.	Crimson Seedless	Effect of CPPU (0, 2, 3, and 4 mg/L) on berry size	CPPU (2, 3, and 4 mg/L) increased berry diameter	[242]
Table grape *Vitis vinifera* L.	Thompson Seedless	Effect of CPPU (0, 20, and 40 mg/L) and GA_3_ (40 mg/L) on fruit quality	CPPU (40 mg/L) + GA_3_ significantly increased diameter, length, L/D ratio, and volume	[208]
Table grapes *Vitis vinifera* L.	Flame Seedless	Effect of CPPU (0 and 20 g/ha) and ABA (0, 300, and 600 mg/L) on berry weight, length, diameter, firmness of harvested fruits in 2005. In addition, the color characteristics (lightness, chroma, and hue) were determined	When compared to the control, CPPU increased the berry weight, length, diameter, firmness, and SSC. All the color characteristics were improved in CPPU-treated fruits	[243]
Table grapes *Vitis vinifera* L.	Flame Seedless	Effect of CPPU (0, 5, 10, 15, and 20 g/ha) and ABA (0, 300, and 600 mg/L) on berry weight, length, diameter, and firmness of harvested fruits in 2006. In addition, the color characteristics (lightness, chroma, and hue) were determined	CPPU (15 g/ha) increased berry weight, firmness, and diameter while berry lengths were reduced. Color characteristics were improved in CPPU-treated fruits	[243]
Table grapes *Vitis vinifera* L.	Redglobe (seeded), Santiago	Effect of PGRs (0x GA_3_, 1x GA_3_, CPPU, or 1x GA_3_ + CPPU) on the quality of berries at harvest	All the treatments significantly increased the fruit weight, size, and rachis thickness	[244]
Table grapes *Vitis vinifera* L.	Thompson Seedless (seedless)	Effect of PGRs (2x GA_3_, 8x GA_3_, 2x GA_3_ + CPPU, or 8x GA_3_ + CPPU) on the quality of berries at harvest	Fruit weight, size, and rachis thickness increased in all the treatments	[244]
Table grapes *Vitis vinifera* L.	Redglobe (seeded), Rancagua	Effect of PGRs (0x GA_3_, 1x GA_3_, CPPU, or 1x GA_3_ + CPPU) on the quality of berries at harvest	Treatments increased fruit size and rachis thickness	[244]
Table grapes *Vitis vinifera* L.	Ruby Seedless	Effect of PGRs (2x GA_3_, 8x GA_3_, CPPU, or 8x GA_3_ + CPPU) on the quality of berries at harvest	Treatments improved fruit weight, size, and rachis thickness	[244]
Avocado *Persea americana* Mill.	Fuerte	Effect of preharvest treatments with different cytokinins (10 µM BA and 40 µM TDZ) on fungal decay induced via inoculation with *Colletotrichum gloeosporioides*	Treatment with CKs reduced the level of fungal decay	[245]
Avocado *Persea americana* Mill.	Ettinger	Effect of preharvest treatment with BA (50 µg/mL) on fruit firmness and fungal decay naturally induced by *Colletotrichum gloeosporioides*	BA delayed fruit softening and reduced naturally occurring fruit decay by 17% 14 days after harvest	[245]
Seedless table grapes *Vitis* spp.	Sovereign Coronation	Effect of CPPU concentrations (0, 1, and 10 mg/L) on yield components (weight)	CPPU increased berry weight with increasing concentration	[210]
Seedless table grapes *Vitis* spp.	Summerland Selection 495 grapes	Effect of CPPU concentrations (0, 1, and 10 mg/L) on yield components (weight)	CPPU increased berry weight	[210]
Seedless table grapes *Vitis* spp.	Simone	Effect of CPPU and TDZ concentrations (0, 1, and 10 mg/L) on yield components (weight)	Both CPPU and TDZ significantly increased the berry weight	[210]
Seedless table grapes *Vitis* spp.	Selection 535	Effect of CPPU and TDZ concentrations (0, 1, and 10 mg/L) on yield components (weight)	TDZ significantly increased the berry weight, while CPPU had no influence	[210]

2,4-D = 2,4- dichlorophenoxyacetic acid; ABA = abscisic acid; BA = *N*^6^-benzyladenine; CK = cytokinin; CPPU = *N*-(2-chloro-4-pyridyl)-*N’*-phenylurea; DAFB = days after full bloom; DAFL = days after flowering; DAFS = days after fruit set; DAPF = days after petal fall; DAT = days after treatment; FB = full bloom; GA = gibberellin; L/D = length/diameter ratio; *m*T = *meta*-topolin; NAA = naphthaleneacetic acid; PGRs = plant growth regulators; SSC = soluble solid concentration; TDZ = thidiazuron.

**Table 5 biomolecules-10-01222-t005:** Biochemical and physiological responses in horticultural fruits following preharvest cytokinin application.

Species	Cultivar	Factor(s) Investigated	Major Outcome(s)	Reference
Hardy kiwifruit *Actinidia arguta* (Sieb. et Zucc.) Planch. ex Miq.	Mitsuko	Effect of time (0, 1, 10, and 25 DAPF) application of 5 mg/L CPPU on fruit quality such as TSS and TA	TSS, TA, and ascorbic acid concentrations significantly decreased in CPPU treatments compared with the control	[214]
Hardy kiwifruit *Actinidia arguta* (Sieb. et Zucc.) Planch. ex Miq.	Mitsuko	Effect of CPPU (0, 1, 5, and 10 mg/L) on the fruit quality	TSS, TA, and ascorbic acid content were not enhanced by CPPU	[214]
Kiwifruit *Actinidia deliciosa* (A. Chev.) C.F. Liang and A.R. Ferguson	Hayward with cv. Matua as pollinizer (6:l)	Effect of CPPU (0 and 20 mg/L) on carbohydrate accumulation and metabolism	CPPU-treated fruits had higher soluble sugars than in the control. A higher ADP–glucose pyrophosphorylase activity and chlorophyll content occurred in CPPU treatment	[216]
Pineapple *Ananas comosus* (L.) Merr.	N-67-10	Effect of BA (0, 10, 25, and 50 mg/L) and CPPU (0. 1.25, 2.5, and 5 mg/L) on growth of stem cutting	Production of about 21 buds per stem after treatment with 5 mg/L CPPU for more than 3 h. More than 18 (89%) of the buds eventually sprouted	[220]
Watermelon *Citrullus lanatus* Thunb.	Reina de corazones	Effect of CPPU (0, 50, 100, 150, and 200 mg/L) and 2,4-D (0, 4, 6, 8, and 12 mg/L) on triploid watermelon production and quality	Production and number of fruit obtained with CPPU were similar to bee pollinated treatments. Maximum production was obtained at 100–200 mg/L. CPPU had low sugar accumulation than 2,4-D	[221]
Persimmon fruit *Diospyros kaki* Thunb.	Triumph	Effect of Superlon (mixture of GA_4+7_ and BA; 0 and 40 µg/mL) applied once a month (from 40 DAFS) for three consecutive months on fruit physiological responses	Treatment (40 DAFS) inhibited ethylene and CO_2_ production in the stem end. Treatments (100 DAFS) enhanced cell proliferation under the fruit cuticle. Treated fruits (115 and 190 DAFS) had delayed chlorophyll degradation in both stem and bottom ends of the fruit	[222]
Strawberry *Fragaria × ananassa* Duch.	Akihime	Effect of CPPU (0, 5, 10, and 15 mg/L) preharvest treatment on fruit quality	Low dose of 5 mg/L CPPU increased TSS content, endogenous hormone, which inevitably increased sugar content in fruits in the late developmental stages. 10 mg/L increased TA	[55]
Litchi *Litchi chinensis* Sonn.	Feizixiao	Effect of 0, 10, and 20 days application of ABA (25 mg/L ABA) and CPPU (4 mg/L) on plant physiology	ABA increased anthocyanin levels in fruit pericarp after 20 days, while CPPU reduced the phytochemical content. ABA accelerated chlorophyll degradation in fruits, with CPPU significantly delaying the process	[67]
Apple *Malus domestica* Borkh.	Fuji	Effect of CPPU (10 mg/L), CPPU (10 mg/L) + GA_4+7_ (100 mg/L) on TSS and malic acid at 4 and 30 DAFB	TSS and malic acid content were not significantly different across the treatments relative to the control	[211]
Apple *Malus domestica* Borkh.	Fuji	Effect of different concentrations of CPPU (1, 5, and 10 mg/L) on TSS and malic acid at 10, 20, and 30 DAFB	TSS content significantly reduced for all treatment when compared to the control. Malic acid content were not different across all treatments relative to the control	[211]
Apple *Malus domestica* Borkh.	Oregon Spur Delicious/MM.111	Effect of CPPU (0, 6.25, 12.5, 25, and 50 mg/L) and GA on SSC and TA at FB and 14 DAFB	CPPU applied at FB or 14 DAFB did not affect SSC and TA	[224]
Apple *Malus domestica* Borkh.	Delicious/M.26	Effect of CPPU (0, 5, 10, 15, and 20 mg/L) on fruit quality	CPPU treatment had no significant stimulatory effect on SSC	[224]
Apple *Malus domestica* Borkh.	Empire	Effect of BA (0, 50, 100, and 150 mg/L) and ethephon (0, 50, and 100 mg/L) applied 13 DAFB on nutrient concentration	Foliar K was increased and Ca decreased by BA, while N and Mg were unaffected by both chemicals	[225]
Apple *Malus domestica* Borkh.	Empire	Effect of BA (0, 125, and 250 mg/L), TDZ (0, 62, and 125 mg/L), and ethephon (0, 125, and 250 mg/L) applied 22 DAFB on nutrient concentration	BA increased fruit flesh N and K and decreased P and Ca concentrations	[225]
Apples *Malus domestica* Borkh.	McIntosh	Effect of TDZ (0, 10, or 50 mg/L) on SSC and return bloom at FB and 22 DAFB	SSC decreased with 50 mg/L at 22 DAFB	[226]
Apples *Malus domestica* Borkh.	Empire	Effect of TDZ (0, 1, 5, and 15 mg/L) on SSC at FB and 18 DAFB	SSC increased with TDZ at 18 DAFB	[226]
Apples *Malus domestica* Borkh.	Double Red Delicious	Effect of CPPU, TDZ (0, 5, and 10 mg/L), and promalin (25 mg/L) on fruit quality	Relative to the control, no treatment improved the SSC	[226]
Apples *Malus domestica* Borkh.	McIntosh	Effect of CPPU (0, 1, 2, 4, and 8 mg/L) applied at 5–6 mm on fruit quality	Increasing concentrations of CPPU increased SSC	[227]
Apples *Malus domestica* Borkh.	McIntosh	Effect of application time (petal fall or 5–6 mm) and concentrations (0, 2, 4, or 6 mg/L) of CPPU on fruit characteristics	CPPU caused a linear increase on the fruit flesh, Ca, and SSC	[227]
Apples *Malus domestica* Borkh.	Marshall McIntosh	Effect of BA (0 and 150 mg/L) and ABA (0, 50, 150, 300, and 1000 mg/L) on fruit quality	BA increased SSC and starch rating	[209]
Apples *Malus domestica* Borkh.	Morespur McIntosh	Effect of BA (0, 50, and 100 mg/L) on endogenous CK levels in leaves and fruit	BA increased ZR levels in the fruit (2 DAT) with no effects on the levels of Z. Amount of Z and ZR in the leaves were not affected by BA application	[229]
Mango *Mangifera indica* L.	Kesha	Effect of CPPU (10 and 20 mg/L) and NAA (40 mg/L) on the quality of mango following applications at mustard, pea, and marble stages	Application of CPPU had no significant effect on the quality of fruit in terms of TSS, total sugars, reducing sugars, non-reducing sugars, and acidity	[231]
Mango *Mangifera indica* L.	Alphonso	Effects of different PGRs (e.g., NAA: 20 mg/L, CPPU: 10 mg/L, paclobutrazol: 10 and 25 mg/L) on morpho-physiological response	CPPU resulted in approximately 40–60% increase in total chlorophyll content at the peanut stage	[232]
Pear *Pyrus communis* L.	Akca	Effect of BA (0, 50, 100, and 150 mg/L) and GA_4+7_ (0, 12.5, 25, and 50 mg/L) on fruit quality (SSC, pH, acidity, and fruit color)	All the treatments increased SSC	[205]
Pear *Pyrus pyrifolia* Nakai.	Shuijing	Effect of BA (0 and 1000 μg/mL) and *Cryptococcus laurentii* (1 × 10^8^ cells/mL) alone or in combination on ethylene production, CAT, POD, and LOX activities	Treatments with BA alone or with *C*. *laurentii* increased the CAT activity and an inhibition of POD and LOX activities as well as ethylene production	[252]
Grape vine *Vitis vinifera* L.	Thompson seedless	Effect of preharvest sprays of CPPU (0 and 5 mg/L) on berry quality	CPPU significantly increased juice content when compared to the control	[241]
Grapes *Vitis vinifera* L.	Flame Seedless	Effect of CPPU (0, 1, 2, and 3 mg/L) on TSS, TA, and anthocyanin concentration	CPPU (2 mg/L) increased TA. TSS and anthocyanin concentration were slightly lowered in CPPU treatments than in control	[242]
Grapes *Vitis vinifera* L.	Redglobe	Effect of CPPU (0, 3, 4, and 5 mg/L) on TSS, TA, and anthocyanin concentration	CPPU (5 mg/L) increased TSS when compared to the control. TA and anthocyanin concentration were similar or slightly lowered in CPPU treatments relative to the control	[242]
Grapes *Vitis vinifera* L.	Crimson Seedless	Effect of CPPU (0, 2, 3, and 4 mg/L) on berry size, TSS, TA, and anthocyanin concentration	All concentrations of CPPU increased TA	[242]
Table grapes *Vitis vinifera* L.	Flame Seedless	Effect of CPPU (0 and 20 g/ha) and ABA (0, 300, and 600 mg/L) on SSC, TA and anthocyanins of harvested (2005) fruits	TTA and anthocyanins were reduced by CPPU. All the color characteristics were improved in CPPU-treated fruits	[243]
Table grapes *Vitis vinifera* L.	Flame Seedless	Effect of CPPU (0, 5, 10, 15, and 20 g/ha) and ABA (0, 300, and 600 mg/L) on SSC, TA, and anthocyanins of harvested (2006) fruits	SSC and anthocyanins were reduced. Color characteristics improved in CPPU-treated fruits	[243]
Avocado *Persea americana* Mill.	Fuerte	Effect of different concentration of TDZ (0.9 and 1.8 µM) on epicatechin in callus clones from mesocarp and pericarp	TDZ enhanced the epicatechin content with the highest concentration (212% than in the control) in pericarp treated with 0.9 µM TDZ	[245]
Avocado *Persea americana* Mill.	Fuerte	Effect of different CKs (40 µM BA and 10 µM TDZ) on epicatechin in pericarp 7 and 14 days preharvest of fruits	Both CKs enhanced the epicatechin content with higher concentrations (2-fold increase) observed at 14 days preharvest 10 µM TDZ treatment	[245]
Avocado *Persea americana* Mill.	Fuerte	Effect of TDZ (0.9 µM) on enzyme activity (DFR) in mesocarp	Activity of DFR significantly increased (4-fold) in TDZ-treated mesocarp	[245]
Avocado *Persea americana* Mill.	Fuerte	Effect of different CKs (40 µM BA and 10 µM TDZ) on enzyme activity in pericarp	Both CKs enhanced the activity of F3H and DFR in TDZ and BA treatments, respectively	[245]
Seedless table grapes *Vitis* spp.	Sovereign Coronation	Effect of CPPU concentrations (0, 1, and 10 mg/L) on fruit composition (anthocyanin, Brix, TA, and pH)	TA levels increased with increasing concentration of CPPU, while Brix reduced in the presence of CPPU. Anthocyanin content was highest at 1 mg/L CPPU treatment	[210]
Seedless table grapes *Vitis* spp.	Summerland Selection 495 grapes	Effect of CPPU concentrations (0, 1, and 10 mg/L) on fruit composition (Brix, TA and pH)	Brix decreased in the presence of CPPU, while TA was not affected	[210]
Seedless table grapes *Vitis* spp.	Simone	Effect of CPPU and TDZ concentrations (0, 1, and 10 mg/L) on fruit composition (anthocyanin, Brix, TA and pH)	TDZ significantly increased TA content. Both CPPU and TDZ had no significant effect on Brix. Anthocyanin content was significantly reduced by TDZ treatments while CPPU had no significant effect	[210]
Seedless table grapes *Vitis* spp.	Selection 535	Effect of CPPU and TDZ concentrations (0, 1, and 10 mg/L) on fruit composition (Brix, TA and pH)	CPPU increased Brix and TA. TDZ increased Brix but had no effect on the TA	[210]

2,4-D = 2,4-dichlorophenoxyacetic acid; ABA = abscisic acid; BA = *N*^6^-benzyladenine; Ca = calcium; CAT = catalase; CKs = cytokinins; CO_2_ = carbon dioxide; CPPU = *N*-(2-chloro-4-pyridyl)-*N’*-phenylurea; DAPF = days after petal fall; DAFS = days after fruit set; DAFB = days after full bloom; DFR = dihydroflavanone reductase; F3H = flavanone-3-hydroxylase; FB = full bloom; GA = gibberellins; K = potassium; LOX = lipoxygenase; Mg = magnesium; N = nitrogen; NAA = naphthalene acetic acid; P = phosphorus; POD = peroxidase; TA, titratable acidity; TDZ = thidiazuron; TTA = total titratable acidity; TSS = total soluble solids; SSC = soluble solids concentration; Z = zeatin; ZR = zeatin ribosides.

**Table 6 biomolecules-10-01222-t006:** Effects of preharvest application of cytokinins on postharvest quality of horticultural fruits.

Species	Cultivar	Factor(s) Investigated	Major Outcome(s)	Reference
Kiwifruit *Actinidia deliciosa* C.F.Liang and A.R.Ferguson.	Hayward	Effect of CPPU (0, 10, and 20 mg/L) sprayed at two application times (14 and 35 DAFB) on storage life	CPPU treatment had no positive effect and the storability of fruits was similar to the control	[215]
Kiwifruit *Actinidia deliciosa*	Hayward with cv. Matua as pollinizer (6:l)	Effect of CPPU (0 and 20 mg/L) on postharvest fruit performance (chlorophyll, glucose, fructose, sucrose, and starch content)	After 2.9 weeks of storage, CPPU-treated fruit had higher glucose, fructose, sucrose, and starch content. After 24.3 weeks (6 months) of storage, both control and CPPU-treated fruit had similar amount of glucose, fructose, sucrose, and starch. CPPU application stimulated the content of chlorophyll during storage	[216]
Cucumber *Cucumis sativus* L.	Zaokang	Effects of CPPU, NAA, and GA_4+7_ at 100 mg/L on fruit quality after storage, 10 day	CPPU decreased weight loss, flesh firmness, and had no effect on fruit color. In addition, lower concentrations of phenolic and vitamin C were observed after 10 days of storage	[266]
Persimmon fruit *Diospyros kaki* Thunb.	Triumph	Effect of Superlon (mixture of GA_4+7_ and BA) applied at 40 µg/mL once a month during three consecutive months of storage at 0°C on cracking and incidence of ABS disease	Superlon altered several host responses that affect ABS development. For example, 45% reduction in the cracked area and 40–50% reduction in naturally occurring ABS, while fruit firmness was not affected	[222]
Apple *Malus domestica* Borkh.	Delicious/M.26	Effect of CPPU (0, 5, 10, 15, and 20 mg/L) on fruit quality after 28 weeks storage (0°C)	CPPU-treated fruits were firmer than control treatment	[224]
Apple *Malus domestica* Borkh.	Hi-Early Delicious	Effect of promalin (0 and 25 mg/L) and CPPU (0, 5, 10, and 15 mg/L) on fruit quality after storage	After 7 days of storage at 20 °C, CPPU (5, 10, and 15 mg/L) increased flesh firmness	[224]
Apples *Malus domestica* Borkh.	Double Red Delicious/M.26	Effect of CPPU, TDZ (0, 5 and 10 mg/L), and promalin (0, 25 mg/L) on postharvest storage (fruit firmness) at 0 °C for 26 weeks and fruit quality (senescent breakdown, decay, bitter pit, and cork spot) at 20 °C for 2 weeks	Fruits from the treatments were firmer than the control. No treatment influenced senescent breakdown, decay or bitter pit, and cork spot	[226]
Apples *Malus domestica* Borkh.	Marshall McIntosh/M.9	Effect of CPPU (0 and 8 mg/L) applied over time on flesh firmness, fruit calcium content, and development of storage disorders following air storage at 0°C for 20 weeks	CPPU decreased flesh firmness, increased senescent breakdown, increased scald, and decreased fruit flesh calcium	[227]
Apples *Malus domestica* Borkh.	McIntosh/M.7	Effect of CPPU (0, 1, 2, 4, and 8 mg/L) applied at 5–6 mm on flesh firmness, fruit calcium content, and development of storage disorders following air storage for 20 weeks	Senescent breakdown and scald increased linearly with increasing CPPU concentrations. Fruit firmness and Ca content were not affected by CPPU treatment	[227]
Table grapes *Vitis labrusca* L.	Himrod	Effect of different concentrations (0, 5, 10, and 15 mg/L) of CPPU on berry shatter and rachis necrosis after cold (1°C) storage for 30 days	CPPU treatments reduced berry shatter, 10 days. Rachis necrosis was reduced (33–7% at day 20 and 44–12% at day 30) by 10 mg/L	[240]
Table grapes *Vitis vinifera* L.	Flame Seedless	Effect of CPPU (0, 1, 2, and 3 mg/L) on defects (loose berries, SO_2_ damage, berry crack, decay, soft tissue breakdown, and external bruises) after six weeks (five weeks at –0.5°C plus one week at 7.5°C) of cold storage	CPPU (1, 2, and 3 mg/L) slightly reduced the total loose berries (%). Total cold storage defects (%) were not positively affected by any treatments	[242]
Table grapes *Vitis vinifera* L.	Redglobe	Effect of CPPU (0, 3, 4, and 5 mg/L) on defects (loose berries, SO_2_ damage, berry crack, decay, soft tissue breakdown, and external bruises) after six weeks (five weeks at –0.5°C + one week at 7.5°C) of cold storage	CPPU (5 mg/L) increased the percentage of loose berries. Total cold storage defects (%) were not positively affected by any treatments when compared to the control	[242]
Table grapes *Vitis vinifera* L.	Crimson Seedless	Effect of CPPU (0, 2, 3, and 4 mg/L) on defects (loose berries, SO_2_ damage, berry crack, decay, soft tissue breakdown, and external bruises) after six weeks (five weeks at –0.5 °C + one week at 7.5°C) of cold storage	An increase in CPPU dosage increase the percentage of SO_2_ damage, bruises, and total defects. At 4 mg/L, these increases were significant when compared to the control	[242]
Grape vine *Vitis vinifera* L.	Thompson seedless	Effect of preharvest sprays of CPPU (0 and 5 mg/L), putrescine, GA, salicylic acid, ethephon, ascorbic acid, and calcium chloride on shelf life after 7 days of storage at ambient temperature	CPPU pre-treated fruits were firm. CPPU reduced unmarketable berries (%) and weight loss. CPPU increased berries adherence strength and reduced the percentage of berry shattering	[241]
Table grapes *Vitis vinifera* L.	Redglobe (seeded), Santiago	Effect of PGRs (0x GA_3_, 1x GA_3_, CPPU or 1x GA_3_ + CPPU) on the firmness, shatter, hairline, splitting, hairline plus splitting, bleaching, and gray mold (*Botrytis cinerea*) incidence on berry cheek or at the berry base after 90 days at 0°C + 3 days at 20°C	CPPU had no significant influence on all the investigated parameters after storage period. Treatment with 1x GA_3_ + CPPU influenced the shatter and incidence of gray mold (*Botrytis cinerea*) after storage	[244]
Table grapes *Vitis vinifera* L.	Thompson Seedless (seedless)	Effect of PGRs (2x GA_3_, 8x GA_3_, 2x GA_3_ + CPPU, or 8x GA_3_ + CPPU) on the firmness, shatter, hairline, splitting, hairline plus splitting, bleaching, and gray mold (*Botrytis cinerea*) incidence on berry cheek or at the berry base after 60 days at 0 °C + 3 days at 20 °C	All treatments had a significant effect on incidences of shatter, hairline, splitting, and gray mold on grapes after the storage period	[244]
Table grapes *Vitis vinifera* L.	Redglobe (seeded), Rancagua	Effect of PGRs (0x GA_3_, 1x GA_3_, CPPU or 1x GA_3_ + CPPU) on the firmness, shatter, hairline, splitting, hairline plus splitting, bleaching, and gray mold (*Botrytis cinerea*) incidence on berry cheek or at the berry base after 90 days at 0 °C + 3 days at 20 °C	PGRs had a significant effect on fruit firmness, shatter, bleaching, and gray mold incidence after the storage period	[244]
Table grapes *Vitis vinifera* L.	Ruby Seedless	Effect of PGRs (2x GA_3_, 8x GA_3_, CPPU or 8x GA_3_ + CPPU) on the firmness, shatter, hairline, splitting, hairline plus splitting, bleaching, and gray mold (*Botrytis cinerea*) incidence on berry cheek at the berry base after 60 days at 0 °C + 3 days at 20 °C	PGRs affected fruit firmness, shatter, hairline, and gray mold incidence	[244]

ABS = alternaria black spot; CPPU = *N*-(2-chloro-4-pyridyl)-*N’*-phenylurea; DAFB = days after full bloom; GA = gibberellins/gibberellic acid; NAA = naphthaleneacetic acid; PGRs = plant growth regulators; SO_2_ = sulfur dioxide; TDZ = thidiazuron.

**Table 7 biomolecules-10-01222-t007:** Effects of postharvest application of cytokinins on postharvest quality of horticultural fruits.

Species	Cultivar	Factor(s) Investigated	Major Outcome(s)	Reference
Calamondin *Citrofortunella microcarpa* (Bunge) Wijnands		Effect of BA (0, 1, 10, or 100 mg/L) in delaying de-greening of rind color and fruit quality during short (5 and 9 days) storage under light or dark conditions	BA delayed de-greening of the calamondin fruit in both light and dark conditions. After 5 days, BA had no influence on TSS, TA, sugar content, AA, and organic acid in the fruit juice, while at 9 days, AA content decreased in BA-treated fruits	[254]
Cucumber fruit *Cucumis sativus* L.	Deltastar	Effect of BA (0, 10, 50, and 100 mM) on CI, antioxidant status, and energy status in cucumber at 2 °C storage, 95% relative humidity, temperature-controlled chamber (darkness) for 16 days	BA (50 mM) reduced CI and increased chlorophyll, AA, total phenolics, and total antioxidant levels	[262]
Round summer squash *Cucurbita maxima* var. Zapallito (Carr.) Millan	Zapallito	Effect of BA (0 and 1 mM) on cell wall metabolism, softening, and quality maintenance of refrigerated (5 °C for 0, 13, or 25 days) summer squash	BA delayed wall degradation and softening with higher levels (45%) of tightly bound polyuronides. BA did not affect the color, respiration or sugar–acid balance and prevented phenolic compound accumulation and decreased pectin solubilization	[257]
Litchi *Litchi chinensis* Sonn.		Effects of BA (0.1 g/L, dipping treatment for 10 min) on decay (by inhibiting the growth and development of *Peronophythora litchii*, the major pathogenic fungi) and pericarp browning of harvested litchi fruits in relation to phenolics and ROS metabolism	BA inhibited the decay incidence of harvested litchi, lowered pericarp browning via the reduction in PPO, and increased PAL activity, anthocyanin, and total phenolics. BA reduced H_2_O_2_ and lipid peroxidation, which may account for browning inhibition	[259]
Litchi *Litchi chinensis* Sonn.		Effect of CPPU on pericarp anatomy and the susceptibility to pericarp browning. Fruit-bearing branches at full female bloom were dipped for 10 s in different concentrations of CK (5, 10, and 20 mg/L) at 4 or 7 weeks	Treatment application after 4 weeks enhanced fruit maturity and thickened the pericarp while reducing the rate of postharvest water loss and susceptibility pericarp browning. Seven-week application of treatment increased epidermal cell proliferation	[267]
Apple *Malus domestica* Borkh.	Fuji	Effect of BA (20 µg/mL) and *Cryptococcus laurentii* in reducing the blue mold disease (caused by *Penicillium expansum*) of apple fruit	BA enhanced the efficacy of *Cryptococcus laurentii* in reducing postharvest blue mold disease	[256]
Apple *Malus domestica* Borkh.	Fuji	Effect of BA (20 µg/mL) and *Cryptococcus laurentii* on defense-related enzyme (SOD and POD) activities *in vivo*	SOD activity was induced by BA and *Cryptococcus laurentii*, POD activity did not increase with these treatment when compared to the control	[256]
Apple *Malus domestica* Borkh.	Fuji	Effect of BA (2, 20, 200, and 2000 µg/mL) on population growths of *Cryptococcus laurentii* in apple fruit wounds	The different concentrations of BA had no significant difference in the yeast population during 96 h of incubation	[256]
Olives *Olea europaea* L.	Konservolia/Conservolea	Effect of BA (0 and 75 mg/L) and ethylene (0 and 1000 µl/L) individually or in combination on respiration, ethylene production, color development, and firmness	Ethylene stimulated respiration rates. No positive effect was evident on ethylene production with BA, BA + ethylene treatments. BA stimulated the development of red–purple color in the skin	[268]
Olives *Olea europaea* L.	Konservolia	Effect of BA (0–100 mg/L) on ripening processes of harvested green olives maintained at 12 °C and 25 °C	At both temperatures, BA stimulated color development, ethylene production, and respiration rates but did not affect fruit firmness	[269]
Peach *Prunus persica* L. Batsch.	Hanlumi	Effect of BA (0 and 500 mg/L) on fruit quality (firmness, membrane permeability, TA, and phytochemical content) at 4 °C storage after 16 days as well as the inhibition of *Monilinia fructicola* at low temperature	BA yielded firmer fruits (13.8 vs. 9.38 MPa), protected cell membrane, prevented fruit texture deterioration, and induced specific polyphenol oxidase and peroxidase activities, which triggered stronger host defensive responses	[264]
Peach *Prunus persica* L. Batsch.	Hanlumi	Effect of BA (0, 100, 200, 500, and 1000 mg/L) in reducing brown rot (caused by *Monilinia fructicola*) in vivo at room temperature (25°C)	BA (500 and 1000 mg/L) suppressed disease incidence throughout the incubation period	[264]
Peach *Prunus persica* L. Batsch.	Hanlumi	Effect of BA (0, 100, 500, and 1000 mg/L) in reducing brown rot (caused by *Monilinia fructicola*) in vivo at low temperature (4°C)	No appearance of brown rot was detected at 500 mg/L BA treatment until 336 h and the disease incidence reached 33.3% at the end of experiment	[264]
Peach *Prunus persica* L. Batsch.	Hanlumi	Effect of BA (0 and 500 mg/L) on defense-related enzyme activities in vivo	Higher SOD, POD, and PPO activities were induced in wounds inoculated with BA	[264]
Pear *Pyrus pyrifolia* Nakai.	Shuijing	Effect of BA (0, 2, 20, 200, 500, 1000, and 2000 μg/mL) alone or in combination with the biocontrol yeast *Cryptococcus laurentii* (1 × 10^8^ cells/mL) in controlling blue mold infection (*Penicillium expansum*)	*Cryptococcus laurentii* and BA (1000 μg/mL) were more effective and stable inhibitors of the mold rots. BA (1000 μg/mL) treatments alone or with *Cryptococcus laurentii* increased CAT activity and an inhibited POD and LOX activities as well as ethylene production	[252]

AA = ascorbate/ascorbic acid; BA = *N*^6^-benzyladenine; CAT = catalase; CI = chilling injury; CPPU = *N*-(2-chloro-4-pyridyl)-*N*’-phenylurea; H_2_O_2_ = hydrogen peroxide; LOX = lipoxygenase; PAL = phenyl amonnia lyase; POD = peroxidase; PPO = polyphenol oxidase; ROS = reactive oxygen species; SOD = superoxide dismutase; TA = titratable acidity; TSS = total soluble solids.

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
