# Peer review of "Applications of Cytokinins in Horticultural Fruit Crops: Trends and Future Prospects"

_biomolecules, 2020, doi:10.3390/biom10091222_

Round 1

Reviewer 1 Report

The paper supplies a review on the effects of cytokinin application to horticultural fruiting crops and their propagation.

The review is a good report covering what has been published in the area of horticultural use of cytokinins for horticultural crop production and propagation, including current knowledge published regarding their mode of action.

Having the effects of cytokinin application for micro propagation doesn't totally fit in with the implication given in the title of the review being on fruit crop production and isn't indicated in the keywords either so a literature search for a review on this aspect may not pick up this paper. The propagation being included does form a complete picture but adds a lot to the text and is not really indicated in the title. I would suggest that this aspect is highlighted better or maybe not included.

There are two corrections

Line 218 The reference is cited as author, year  instead of number as per the rest of the text

Line 222 Recent times should be used rather than recent time. or alternatively maybe use recently instead.

Reviewer 2 Report

Review of the Manuscript ID: biomolecules-910118, titled: ”Applications of cytokinins in horticultural fruit crops: trends and future prospects”

     In this review, the authors highlight and critically explore the potential of cytokinins (CKs) in the propagation, growth and general physiology of some fruit crops.
The authors have done a great deal of work in the preparation of this manuscript,  describing the available current information and new knowledge about the molecular mechanisms, and modes of action of CKs to provide a better understanding of their function and regulatory control in horticultural fruit crops. This review also focuses on the application of CKs in pre- and postharvest management practices of horticultural crops.

Therefore, this manuscript is of great scientific and practical importance and I recommend it for publication in Biomolecules (MDPI).

Lines 218, 229, and 655: References (Ni et al., 2017), (Shi et al., 2013) and Ainalidou et al. (2016), should be numbered.
